# Global modeling of aerosol nucleation with a semi-explicit chemical mechanism for highly oxygenated organic molecules (HOMs)

Xinyue Shao[1,2], Minghuai Wang[1,2], Xinyi Dong[1,2], Yaman Liu[1,4], Wenxiang Shen[1,2], Stephen R. Arnold[5], Leighton A. Regayre[5,6,7], Meinrat O. Andreae[8,9], Mira L. Pöhlker[8,10,11], Duseong S. Jo[12], Yue Man[1,4], and Ken S. Carslaw[5]

[1]School of Atmospheric Science, Nanjing University, Nanjing, 210023, China
[2]Joint International Research Laboratory of Atmospheric and Earth System Sciences & Institute for Climate and Global Change Research, Nanjing University, Nanjing, 210023, China
[3]Frontiers Science Center for Critical Earth Material Cycling, Nanjing University, Nanjing, China
[4]Zhejiang Institute of Meteorological Sciences, Hangzhou, 310008, China
[5]Institute for Climate and Atmospheric Science, School of Earth and Environment, University of Leeds, Leeds, LS2 9JT, UK
[6]Met Office Hadley Centre, Exeter, Fitzroy Road, Exeter, Devon, EX1 3PB, UK
[7]Centre for Environmental Modelling and Computation, School of Earth and Environment, University of Leeds, Leeds, LS2 9JT, UK
[8]Max Planck Institute for Chemistry, Mainz, 55020, Germany
[9]Scripps Institution of Oceanography, University of California San Diego, La Jolla, USA
[10]Leipzig Institute for Meteorology, Universität Leipzig, 04103 Leipzig, Germany
[11]Experimental Aerosol and Cloud Microphysics Department, Leibniz Institute for Tropospheric Research, 04318 Leipzig, Germany
[12]Atmospheric Chemistry Observations and Modeling Laboratory, National Center for Atmospheric Research, Boulder, CO 80103, USA

*Correspondence to*: Minghuai Wang (minghuai.wang@nju.edu.cn), Xinyi Dong (dongxy@nju.edu.cn)

**Abstract.** New particle formation (NPF) involving organic compounds has been identified as an important process affecting aerosol particle number concentrations in the global atmosphere. Laboratory studies have shown that highly oxygenated organic molecules (HOMs) can make a substantial contribution to NPF, but there is a lack of global model studies of NPF with detailed HOMs chemistry. Here, we incorporate a state-of-the-art biogenic HOMs chemistry scheme with 96 chemical reactions to a global chemistry-climate model and quantify the contribution to global aerosols through HOMs-driven NPF. The updated model captures the frequency of NPF events observed at continental surface sites (normalized mean bias changes from -96% to -15%) and shows reasonable agreement with measured rates of NPF and sub-20 nm particle growth. Sensitivity simulations show that compared to turning off the organic nucleation rate, turning off organic initial growth results in a more substantial decrease in aerosol number concentrations. Globally, organics contribute around 45% of the annual mean vertically-integrated nucleation rate (at 1.7 nm) and 25% of the vertically-averaged growth rate. The inclusion of HOMs-related processes leads to a 39% increase in the global annual mean aerosol number burden and a 33% increase in cloud condensation nuclei (CCN) burden at 0.5% supersaturation compared to a simulation with only inorganic nucleation. Our work predicts a greater contribution of organic nucleation to NPF than previous studies due to the semi-explicit HOMs mechanism and an updated

inorganic NPF scheme. The large contribution of biogenic HOMs to NPF on a global scale could make aerosol sensitive to changes in biogenic emissions.

## 1 Introduction

Aerosols exert multifaceted impacts on both climate and human health across a range of environments (Wang and Penner, 2009; Rosenfeld et al., 2014; Shiraiwa et al., 2017; Bellouin et al., 2020; Carslaw, 2022; Rosenfeld et al., 2008). Atmospheric new particle formation (NPF) is a significant contributor to aerosol number concentration, involves the nucleation of stable molecular clusters via gas-to-particle conversion and their subsequent growth through the condensation of precursor vapours (Merikanto et al., 2009; Spracklen et al., 2010; Kerminen et al., 2018; Kulmala, 2003). While both the nucleation and growth of new particles are commonly linked to sulfuric acid ($H_2SO_4$) owing to its low volatility, $H_2SO_4$ and its inorganic clusters (e.g. $H_2SO_4$-$NH_3$ clusters) alone are insufficient to explain the rapid nucleation rates observed in forested regions minimally affected by anthropogenic pollution (Kuang et al., 2008; Sihto et al., 2006; Kerminen et al., 2018; Stolzenburg et al., 2020; Weber et al., 1997; Boy et al., 2008; Paasonen et al., 2010; Andreae et al., 2022). Furthermore, while $H_2SO_4$ frequently initiates cluster formation, its concentration does not account for the high growth rates of particles larger than 3 nm diameter (Ehn et al., 2014; Deng et al., 2020).

Laboratory studies and ambient measurements have shown that highly oxygenated organic molecules (HOMs) largely account for the particle nucleation and growth rate in forested areas owing to their extremely low volatility. Riccobono et al. (2014) revealed a nucleation mechanism involving both $H_2SO_4$ and oxidized organic molecules from the very first step, and including this mechanism in a global aerosol model yielded a seasonal cycle of particle concentrations in the continental boundary layer, in good agreement with observations. Jokinen et al. (2015) found monoterpene-derived HOMs promote NPF under continental conditions using chamber experiments. Kirkby et al. (2016) showed that the rate of formation of particles from biogenic HOMs, in the absence of $H_2SO_4$, can be enhanced by 1–2 orders of magnitude by ions based on CERN CLOUD (Organisation Européene pour la Recherche Nucléaire - Cosmics Leaving Outdoor Droplets) experiments. In addition to contributing to the formation of 1-2 nm clusters, Ehn et al. (2014) showed HOMs made important contributions to the particle growth with diameters between 5 and 50 nm in northern forests, which was recently explained by Mohr et al. (2019) at the molecular level. Bianchi et al. (2016) also showed observational evidence that NPF occurs mainly through condensation of HOMs at high altitude.

Although HOMs are necessary for NPF in the absence of $H_2SO_4$ (Kirkby et al., 2016), the molecular structures and formation pathways of HOMs remain uncertain and are treated in a variety of ways in models. Gordon et al. (2016) simulated monoterpene-derived HOMs formation using an empirical or semiempirical fixed yield of HOMs from the first stage monoterpene

oxidation products, although with highly simplified HOMs chemistry. Zhu et al. (2019) added some explicit chemical mechanisms for HOMs, but they did not consider autoxidation and used a less stringent definition of HOMs than recommended in Bianchi et al. (2019). Roldin et al. (2019) used a more explicit reaction mechanism to treat the generation of HOMs through autoxidation and cross-reactions of α-pinene oxidation products in a 1-D column model. Weber et al. (2020) used a similarly explicit mechanism over the boreal forest in Finland and the southeast USA, although not on a global scale.

There is thus a lack of global-scale simulations of NPF with explicit HOMs chemistry and quantification of the contribution of organics to aerosol and CCN number concentrations. Recently, Xu et al. (2022) summarized the various chemical mechanisms of HOMs, including monoterpene-derived peroxy radical (MT-RO$_2$) unimolecular autoxidation and self- and cross-reactions with other RO$_2$ species, and evaluated them in the GEOS-Chem global model. However, they did not quantify the effects of HOMs participating in NPF. Here, we incorporate the representation of HOMs from Xu et al. (2022) within a global chemistry-climate model and then quantify the contribution of HOMs to aerosol number concentration globally. Inorganic nucleation rates involving H$_2$SO$_4$ and NH$_3$ as well as ion-induced pathways based on the CLOUD chamber experiments are also included (Dunne et al., 2016), replacing a simpler scheme based on H$_2$SO$_4$ and NH$_3$ (Vehkamaki et al., 2002; Merikanto et al., 2007).

The model and field measurements used in this study are documented in Section 2. Section 3 evaluates outputs of the updated model, including nucleation and growth rates, frequencies of NPF events, and aerosol number concentrations. Additionally, four sensitivity experiments aimed at investigating uncertainties in concentrations of organic nucleating species and disentangling the roles of nucleation and growth processes are conducted. Section 4 quantifies the contributions of organic-related processes to nucleation rate, growth rate, aerosol, and CCN number concentrations globally. Section 5 compares the proportion of the organic nucleation rate with previous studies. Results are summarized and discussed in Section 6.

## 2 Data and methods

### 2.1 Model configuration

We use the atmospheric component of the Community Earth System Model (CESM) version 2.1.0, the Community Atmosphere Model version 6, augmented with comprehensive tropospheric and stratospheric chemistry (CAM6-Chem) (Emmons et al., 2020). Biogenic emissions are dynamically simulated using the Model of Emissions of Gases and Aerosol from Nature version 2.1 (MEGAN2.1) (Guenther et al., 2012). We use the historical anthropogenic emissions developed by the Community Emission Data System (CEDS v2017-05-18) in support of phase 6 of the Coupled Model Intercomparison Project (CMIP6) (Hoesly et al., 2018). Monthly biomass burning emissions are from the historical global biomass burning emissions inventory for CMIP6 (van Marle et al., 2017). Emissions for the 1997 to 2015 period in this inventory have been

derived from satellite-based emissions from the Global Fire Emissions Database (van der Werf et al., 2017). The vertical distribution of biomass burning emissions is taken from Dentener et al. (2006). All the emission data can be downloaded from: https://svn-ccsm-inputdata.cgd.ucar.edu/trunk/inputdata/atm/cam/chem/emis/. The previous study (Paulot et al., 2018) reported that there is an overestimation of $SO_2$ emissions over China after 2007 due to the underestimation of the $SO_2$ reduction. Therefore, emissions in China were replaced by the multi-resolution emission inventory for China (MEIC) (http://www.meicmodel.org) (Li et al., 2017; Yue et al., 2023) which considerably improves Chinese emission inventories, compared to the earlier large-scale studies (Zheng et al., 2009; Zhou et al., 2017). CAM6-Chem utilizes a four-mode version of the Modal Aerosol Module (MAM4) (Liu et al., 2016), coupled with the Model for Simulating Aerosol Interactions and Chemistry (MOSAIC) (Zaveri et al., 2021) to explicitly represent the heterogeneous uptake of isoprene-epoxydiols (IEPOX) onto sulfate aerosols and subsequent production of isoprene-epoxydiols (Jo et al., 2019; 2021). Following Liu et al. (2023), we adopt a modest photolysis rate for monoterpene-derived secondary organic aerosols, constituting 2.0% of the $NO_2$ photolysis frequency (Bianchi et al., 2019; Krapf et al., 2016; Zawadowicz et al., 2020). All simulations were run at a horizontal resolution of 0.95° latitude and 1.25° longitude, with a vertical resolution extending up to approximately 40 km across 32 layers (Emmons et al., 2020). To follow the observed meteorological conditions and initialize realistic meteorological conditions, meteorological fields (temperature and wind profiles, surface pressure, surface stress, surface heat and moisture fluxes) are nudged toward Modern-Era Retrospective analysis for Research and Applications (MERRA2) reanalysis (Kooperman et al., 2012), which allows model-observation comparisons that are unaffected by variability in synoptic-scale model dynamics. We evaluate model performance against observations from multiple years (Section 3), where in each case the anthropogenic emissions and model meteorology correspond to values associated with the observation year.

We incorporate advanced chemical reactions involving the formation of HOMs, since in the default configuration of CAM6-Chem, organics are not involved in either the nucleation or the initial growth processes of aerosols (sub-20 nm). These include MT-$RO_2$ unimolecular H-shifts (i.e., "autoxidation") and self- and cross-reactions with other $RO_2$ species, guided by laboratory-derived mechanistic parameters from Xu et al. (2022). In total, 24 reactions in CAM6-Chem were modified and 96 reactions were added. Descriptions of chemical mechanisms are shown in Text S1, and the final products, including HOMs and accretion products (ACC, C15 and C20), are summarized in Table 1.

**Table 1. The molecular formula, saturated vapor concentration (C*) at 300K, enthalpy of vaporization (ΔH$_{vap}$) and corresponding volatility class of newly added organics.**

| Species | Short Name | Molecular formula | log(C*) (μg m$^{-3}$) | ΔH$_{vap}$ (kJ mol$^{-1}$) | Volatility Bin[a] |
|---------|-----------|-------------------|----------------------|---------------------------|------------------|
| HOMs | C10-NON [b] | $C_{10}H_{14}O_9$ | -3.22 | 164.0 | LVOC |

| | C10-ON[c] | $C_{10}H_{14}O_9N$ | -3.31 | 164.0 | LVOC |
|---|---|---|---|---|---|
| ACC[d] | C15 | $C_{15}H_{18}O_9$ | -5.20 | 186.0 | ELVOC |
| | C20 | $C_{20}H_{32}O_8$ | -9.53 | 230.0 | ULVOC |

[a] LVOC, ELVOC and ULVOC represent low/ extreme-low/ ultra-low volatility organic compounds, respectively.

[b] NON represents non-organonitrates.

[c] ON represents organonitrates.

[d] ACC represents accretion products.

## 2.2 Nucleation and growth scheme in CAM6-Chem

### 2.2.1 Nucleation scheme in default CAM6-Chem

The default configuration of CAM6-Chem (Default, Table 2) includes binary homogeneous nucleation of $H_2SO_4$-$H_2O$ (Vehkamaki et al., 2002) and ternary homogeneous nucleation of $H_2SO_4$-$NH_3$-$H_2O$ (Merikanto et al., 2007). Additionally, within the boundary layer the model includes the empirical mechanism of Kulmala et al. (2006) and Sihto et al. (2006) as first used in a global model by Spracklen et al. (2006):

$$j_{1nm} = A \, [H_2SO_4] \tag{1}$$

where A ($1.0^{-6} \, \mathrm{s^{-1}}$) is the rate constant chosen from the median values derived in case studies (Sihto et al., 2006).

### 2.2.1 Updated inorganic nucleation scheme

Most existing models tend to overestimate the sensitivity of the nucleation rate to sulfuric acid concentrations when relying solely on classical nucleation theories of sulfuric acid (Ehn et al., 2014; Mann et al., 2014; Scott et al., 2014). Therefore our study updates the inorganic nucleation parameterizations in CAM6-Chem, drawing upon data from the CLOUD chamber experiments (Kirkby et al., 2016; Dunne et al., 2016). The updated schemes incorporate $H_2SO_4$, $NH_3$ and ions. The inorganic NPF rates at a mobility equivalent diameter of 1.7 nm are calculated by summing the following rates (Dunne et al., 2016):

1. Binary neutral (indicated by b,n, $J_{SA}$) and ion-induced (b,i, $J_{SA,i}$) NPF involving sulfuric acid and water:

$$J_{SA} = K_{b,n}(T)[H_2SO_4]^{P_{b,n}} \tag{2}$$

$$J_{SA,i} = K_{b,i}(T)[H_2SO_4]^{P_{b,i}} \, [n-] \tag{3}$$

$K(T)$ are temperature-dependent prefactors, $P_i$ are constant parameters, and $[n-]$ is the concentration of negative ions produced from galactic cosmic rays (equal to $[n\pm]$ in Eq. (8), which is parameterized in the Text S2).

2. Ternary neutral (indicated by t,n, $J_{SA-NH3}$) and ion-induced (t,i, $J_{SA-NH3,i}$) NPF involving sulfuric acid, ammonia and water:

$$J_{SA-NH3} = K_{t,n}(T)f_n([NH_3], [H_2SO_4])[H_2SO_4]^{P_{t,n}} \tag{4}$$

$$J_{\text{SA-NH3,i}} = K_{t,i}(T) f_i([\text{NH}_3], [\text{H}_2\text{SO}_4])[\text{H}_2\text{SO}_4]^{P_{t,i}} [n-] \tag{5}$$

where the $f([\text{NH}_3], [\text{H}_2\text{SO}_4])$ are functions of the ammonia and sulfuric acid gas phase concentrations, also involving free-fitting parameters.

### 2.2.2 New organic nucleation scheme

There is no organic nucleation scheme in the Default (Table 2) so organic NPF rates at a 1.7 nm mobility equivalent diameter were included as the sum of the following parameterizations:

1. The rate of heteromolecular nucleation of sulfuric acid and organics (HET, $J_{\text{SA-Org}}$) is parameterized following Riccobono et al. (2014), depending on both $\text{H}_2\text{SO}_4$ and organic nucleating species concentration:

$$J_{\text{SA-Org}} = K_m [\text{H}_2\text{SO}_4]^2 [\text{HOM} + \text{ACC}] \tag{6}$$

where ACC are accretion products (Table 1) and $K_m$ is the multicomponent prefactor, which equals to $3.27 \times 10^{-21}$ cm$^6$ s$^{-1}$ (Riccobono et al., 2014).

2. The rate of neutral pure organic nucleation (NON, $J_{\text{Org,n}}$) and ion-induced pure organic nucleation (ION, $J_{\text{Org,i}}$) are param-
eterized based on Kirkby et al. (2016):

$$J_{\text{Org,n}} = a_1 [\text{ACC}]^{a_2 + \frac{a_5}{[\text{ACC}]}} \tag{7}$$

$$J_{\text{Org,i}} = [n \pm] a_3 [\text{ACC}]^{a_4 + \frac{a_5}{[\text{ACC}]}} \tag{8}$$

where ACC are in units of $10^7$ molecules cm$^{-3}$, parameters $a_n$ are determined from fits to experimental data (Dunne et al., 2016), and $[n\pm]$ is the ion concentration produced from galactic cosmic rays (Text S2). A temperature dependence for the
organic nucleation rates was introduced by multiplying by exp(-($T$-278)/10) as suggested in Dunne et al. (2016).

### 2.2.3 Updated particle growth scheme

The growth rate of nuclei is important for the survival probability up to larger sizes and eventually contribution to CCN (Pierce and Adams, 2009; McMurry et al., 2005). The effective production rate of 20 nm diameter particles (the smallest size simulated by the model) is calculated using the Kerminen and Kulmala (2002) formula:

$$j_{20\text{nm}} = j_{1.7\text{ nm}} \exp\left[-\left(\frac{1}{1.7} - \frac{1}{20}\right)\frac{\gamma \text{CS}'}{\text{GR}}\right] \tag{9}$$

where CS' is the reduced (simplified) condensation sink, $\gamma$ is a proportionality factor, and GR is the growth rate. The reduced (simplified) condensation sink (CS') is calculated as (Kerminen and Kulmala, 2002):

$$\text{CS}' = \frac{\text{CS}}{4\pi D_i} \tag{10}$$

where CS is the condensation sink and $D_i$ is the vapor diffusion coefficient. CS′ largely depends on CS and represents the surface area of preexisting aerosols.

In the Default simulation, sub-20 nm particle growth is solely caused by condensation of $H_2SO_4$ and is approximated by Kerminen and Kulmala (2002):

$$GR = \frac{3.0 \times 10^{-9}}{\rho} v_{H2SO4} M_{H2SO4} c_{H2SO4} \tag{11}$$

where $v_{H2SO4}$ is the mean molecular speed of $H_2SO_4$, $M_{H2SO4}$ is the molecular weight of $H_2SO_4$, $c_{H2SO4}$ is the gas phase concentration of $H_2SO_4$, and $\rho$ is the density of the nuclei. $3.0 \times 10^{-9}$ is an approximation of the product of many parameters (Kerminen and Kulmala, 2002).

Neglecting organic vapor condensation on sub-20 nm particles will lead to insufficient growth rates and potentially reduced survival of newly formed particles (Pierce and Adams, 2009). Therefore, the condensation of monoterpene-derived condensable organic compounds (COC), including HOMs and ACC, to newly formed particles is added in our updated model. The enhanced growth rate of particles from 1 nm to 20 nm is then parameterized as follows:

$$GR = \frac{3.0 \times 10^{-9}}{\rho} (v_{H2SO4} \times M_{H2SO4} \times c_{H2SO4} + v_{COC} \times M_{COC} \times [c_{COC} - c^*_{COC}]) \tag{12}$$

where $v_{COC}$ is the mean molecular speed of COC (ACC and HOMs), $M_{COC}$ is the molecular weight of COC, $c_{COC}$ is the gas phase concentration of COC, and $c^*_{COC}$ is saturated vapor concentration of COC which is parameterized in Text S3.

Simulations with the updated inorganic nucleation scheme (i.e., Eq. (2)-(5)) are named "Inorg" and simulations including also the new organic mechanisms (i.e., Eq. (6)-(8) and (12)) are named "Inorg_Org" (Table 2).

**2.3 Method of evaluating NPF-related variables**

In addition to evaluating aerosol concentrations, we also evaluate NPF event properties in terms of the nucleation rate, growth rate and frequency of occurrence of NPF events. CAM6-Chem does not incorporate a nucleation mode, so we employ a threshold of $j_{20nm}$ (Eq. (9)) to define the occurrence of NPF events (i.e., when $j_{20nm}$ > threshold). Then we could evaluate the NPF frequency (fraction of days) by defining an "NPF day" as a day during which $j_{20nm}$ is higher than a threshold value. Also, the method to evaluate these NPF properties during "NPF day" is described in Text S4.

CAM6-Chem utilizes a four-mode version of the Modal Aerosol Module (MAM4) (Liu et al., 2016), including Aitken mode (with diameter 9 to 52 nm), accumulation mode (54 to 480 nm), coarse mode (400 to 40000 nm), and primary mode (10 to 100 nm). The integral concentration from 0 to $r_p$ is computed using the error function (erf):

$$N_{>r_p} = N_{mode} \left( \frac{1}{2} + \frac{1}{2} \text{erf} \left( \frac{x}{\sqrt{2}} \right) \right) \qquad (13)$$

where x is defined as:

$$x = \frac{\ln(r_p/r_m)}{\ln\sigma} \qquad (14)$$

where $\sigma$ is the geometric standard deviation (the width) of the lognormal distribution and $r_m$ is the median radius of the mode. The integral concentration above $r_p$ is therefore calculated as:

$$N_{>r_p} = N_{mode} - N_{<r_p} \qquad (15)$$

The temporal frequency of the nucleation rate, growth rate, and condensation sink written out of the model is hourly, and the
220 simulation periods are consistent with the observation period (with an additional month for spin-up). For aerosol number concentrations (including over oceans and land), the model outputs data on a monthly basis, and we compare these monthly averages with observations. When comparing the aerosol and CCN number concentrations with the field campaign in the Amazon basin, the output frequency from the model is hourly. Then, we slice the aircraft measurements of aerosol and CCN number concentrations vertical profiles according to the model output dimensions (four dimensions including time, height,
latitude and longitude). We average all measurement data within each slice and compare it with the corresponding model output data.

**2.4 Sensitivity experiments**

We performed two simulations to quantify the relative contributions of the nucleation rate (Only_NR) and growth rate
(Only_GR) to aerosol concentrations in order to separate the contribution of the organic compounds to each of these processes. The Only_NR and Only_GR simulations employ the same settings as Inorg_Org (Table 2), but in Only_NR, the organic-involved particle growth is disabled (i.e., Eq. (11) is used instead) and in Only_GR, the organic-involved nucleation rates (i.e., Eq. (6)-(8)) are disabled.

We also conducted two sensitivity simulations to examine uncertainties in concentrations of HOMs (Table 2): sensitivity to
235 the branching ratio from the first generation of monoterpene (MT) reactions with $O_3$/OH that can be auto-oxidized (Low_Br), and sensitivity to the rate of termination reaction involving NO (Slow_NO), sensitivity to the autoxidation temperature dependence (High_temp and Low_temp), sensitivity to the autoxidation rate (Fast_auto and Slow_auto) and sensitivity to the self-/cross-reaction rate (Slow_accr) (Table 2). In Inorg_Org, the branching ratios for the MT-derived peroxyl radicals (MT-$RO_2$), which can be further auto-oxidized are set at 80% for MT+$O_3$ and 97% for MT+OH reactions, corresponding to the high
values reported in Xu et al. (2022). In the Low_Br simulation (Table 2), the branching ratio for MT-$RO_2$ is set as 25% for MT

+ O$_3$ and 92% and MT + OH. Both the high and low branching ratios fall within the range of previous studies (Lee et al., 2023; Pye et al., 2019; Weber et al., 2020; Xu et al., 2018; Jokinen et al., 2015; Roldin et al., 2019). In Slow_NO, the reaction rate of MT-HOM-RO$_2$+NO (MT-HOM-RO$_2$, the second-generation product of autoxidation (Text S1) is set as one-fifth of that in Inorg_Org, given that the simulated NO concentration is fourfold higher than the measured values in the boreal forest in Finland and in the southeast USA (Fig. S3 and S2 in Liu et al. (2024)). In High_temp and Low_temp, the temperature depend­ence of the autoxidation rate is set to lower and upper limits (i.e. representing the possible higher and lower bound of activation energy, Table S8) based on chamber experiments (Roldin et al., 2019; Weber et al., 2020). In Fast_auto and Slow_auto, the autoxidation reaction rates are multiplied by 10 and 0.1 respectively. In Slow_accr, the rate of self-/cross- reactions are set as the lower value (Table S9) based on chamber experiments (Weber et al., 2020; Berndt et al., 2018).

**Table 2. Configurations of CESM2.1.0 Experiments**

| Test Name | Updated inor­ganic nuclea­tion | HOMs chemistry | Organic Nucleation | Organic Growth | Other Changes |
|---|---|---|---|---|---|
| Default | × | × | × | × | / |
| Inorg | ✓ | ✓ | × | × | / |
| Inorg_Org | ✓ | ✓ | ✓ | ✓ | / |
| Only_NR | ✓ | ✓ | ✓ | × | / |
| Only_GR | ✓ | ✓ | × | ✓ | / |
| Low_Br | ✓ | ✓ | ✓ | ✓ | Lower branch ratio of the first generation prod­uct (MT-RO$_2$) from MT + O$_3$ and MT + OH, which could be further auto-oxidized |
| Slow_NO | ✓ | ✓ | ✓ | ✓ | Rate of MT-HOM-RO$_2$ + NO generating HOMs, multiplied by 0.2 |
| High_temp | ✓ | ✓ | ✓ | ✓ | Autoxidation rate with high temperature depend­ence (Roldin et al., 2019) (Table S8) |
| Low_temp | ✓ | ✓ | ✓ | ✓ | Autoxidation rate with low temperature depend­ence (Weber et al., 2020) (Table S8) |
| Fast_auto | ✓ | ✓ | ✓ | ✓ | Autoxidation rate multiplied by 10 |
| Slow_auto | ✓ | ✓ | ✓ | ✓ | Autoxidation rate multiplied by 0.1 |
| Slow_accr | ✓ | ✓ | ✓ | ✓ | Using slower self-/cross reaction rate derived from Weber et al. (2020) and Berndt et al. (2018) (Table S9) |

## 2.5 Observation data

Observational data used in this study are from ships, stations, and aircraft (see Table 3). Measurements from the Canadian
Aerosol Baseline Measurement Program (CABM), the Reactive Halogens in the Marine Boundary Layer (RHaMBLe), and
Aerosol-Cloud Coupling and Climate Interactions in the Arctic (ACCACIA) are compared with simulated N10 or N20 (number concentrations for particles with diameters larger than 10 nm or 20 nm), since these two variables are most sensitive to
aerosol nucleation and initial growth. European Aerosol Cloud Climate and Air Quality Interactions projects (EUSAAR-EU-CAARI) (Asmi et al., 2011; Kulmala et al., 2009) provide measured N30 and N50 (number concentrations for particles with
diameters larger than 30 nm and 50 nm respectively), and these larger particles are more important for the condensation sink
(CS) of HOMs and other precursor vapors during NPF. All of the above-mentioned data were processed in the Global Aerosol
Synthesis and Science Project (GASSP) (Reddington et al., 2017). Measured N20 and CCN concentrations at 0.5% supersaturation from the aircraft campaign Aerosol, Cloud, Precipitation, and Radiation Interactions and Dynamics of Convective
Cloud Systems (ACRIDICON–CHUVA) (Wendisch et al., 2016) are used to examine the effect of the inclusion of organic
NPF on the profile of CCN concentrations in an organic-dominated tropical environment. We also use ground station measurements of nucleation rates, growth rates, CS, and NPF frequencies during specific time periods that correspond to the simulations. Full information of stations is listed in Tables S1 and S2.

**Table 3. Field measurements used in this study**

| Campaign | Platform | Dates | Region | Variables |
|---|---|---|---|---|
| RHaMBLe (http://www.cas.manchester.ac.uk/re-sprojects/rhamble/cruise/) | Ship | 17 May – 9 June 2007 | North Atlantic Ocean (-25.05° W – 8.35° W; 16.32 – 46.14° N) | N10 |
| ACCACIA (http://arcticaccacia.wordpress.com) | Ship | 12 July – 13 August 2013 | Arctic between Norway and Svalbard (20.70 – 34.84° E; 55.73 – 83.32° N) | N10 |
| CABM (https://ec.gc.ca/air-sc-r) | Station | 23 October 2012 – 1 January 2013 | Ellesmere Island, Canada (62.34° W, 82.49° N) Egbert, Canada (79.78° W, 44.23° N) | N20 |
| EUSAAR-EUCAARI (Asmi et al., 2011) | Station | 1 January 2008 – 1 January 2010 | Europe | N30, N50 |
| ACRIDICON–CHUVA (Andreae et al., 2018) | Aircraft | September 2014 | Amazon Basin | N20 CCN (0.5%ss) |

## 3 Evaluation of the updated NPF scheme

In this section, we evaluate the results derived from the updated model (Inorg_Org) and focus on the comparison between Inorg_Org and Inorg (definitions in Table 2). Specifically, we compare the nucleation rate, sub-20 nm particle growth rate, NPF event frequency, and condensation sink (CS) (Fig. 1) between simulations and measurements. Results from sensitivity tests (Low_Br and Slow_NO) are used to evaluate the effect of uncertainties in HOMs chemistry on aerosol (Figs. 2 and 3) and CCN concentrations (Fig. 4).

### 3.1 Evaluation of NPF-related variables

The properties of the nucleation events themselves (formation rates, growth rates, and event frequencies) provide the best test of NPF schemes, while state variables like particle concentration have many other sources of error in a model.

As shown in Fig. 1a, at most grounds stations, the nucleation rate in Inorg_Org agrees better with measurements than Inorg (normalized mean bias, NMB changes from -97% in Inorg to -64% in Inorg_Org). The improvement is particularly clear in non-urban areas where biogenic organic nucleation plays a substantial role, such as Hyytiälä, Ozark Forest, Po Valley, and Leicester (NMB changes from -92% in Inorg to -34% in Inorg_Org, Fig. 1a). In these regions, the nucleation rate increases by at least a factor of 8 when the organic nucleation mechanisms are included (Inorg_Org compared to Inorg). In Toronto and Gadanki, the nucleation rate becomes detectable following the incorporation of organic nucleation mechanisms, in good agreement with observations (9.2 $cm^{-3}$ $s^{-1}$ in Inorg_Org compared to 12.9 $cm^{-3}$ $s^{-1}$ measurement of Toronto; 1.6 $cm^{-3}$ $s^{-1}$ in Inorg_Org compared to 1.2 $cm^{-3}$ $s^{-1}$ in measurement of Gadanki, Table S4). However, in multiple urban regions of China, the nucleation rate remains underestimated (NMB > -50%). This is likely because the effects of anthropogenic-derived HOMs and amines are not accounted for in this study, and these effects will be strongest in urban regions. Hong Kong serves as a stark example, where the nucleation rate shows minimal change when the biogenic-organic nucleation scheme is implemented, rising slightly from 0.3 $cm^{-3}$ $s^{-1}$ (Inorg) to 0.31 $cm^{-3}$ $s^{-1}$ (Inorg_Org). Several other Chinese megacities, including Beijing and Nanjing, show similar behaviour.

Figure 1b shows that the growth rate in Inorg is underestimated (NMB = -54%) but is overestimated at most sites in Inorg_Org (NMB = 39%). The underestimation of the sub-20 nm growth rate in Inorg is due to an almost zero nucleation rate at around 1 nm. Consequently, the absence of a nucleation rate results in the absence of NPF events and thus a zero growth rate. In contrast, in Inorg_Org, the NPF frequency is simulated accurately compared to Inorg (Fig. 1c). One contributing factor to the overestimation of the growth rate in Inorg_Org is the overestimation of the $H_2SO_4$ concentration, a feature of CAM6, as evidenced by comparisons with previous model simulations (Table S6) and measurements (Table S7). This discrepancy is particularly noticeable in China, where $H_2SO_4$ dominates the growth rate (further discussed in Fig. 7, Section 4). This is also sup-

ported by overestimated growth rates in Beijing, Qingdao, and Hongkong in Inorg, which considers only the $H_2SO_4$ contribution to sub-20 nm growth. This suggests excessive $H_2SO_4$ is a feature of the default model. The growth rate of new particles in Hong Kong is zero (Fig. 1b) since there are almost no newly formed particles (nucleation rate ~ 0 $cm^{-3}$ $s^{-1}$, Fig. 1a).

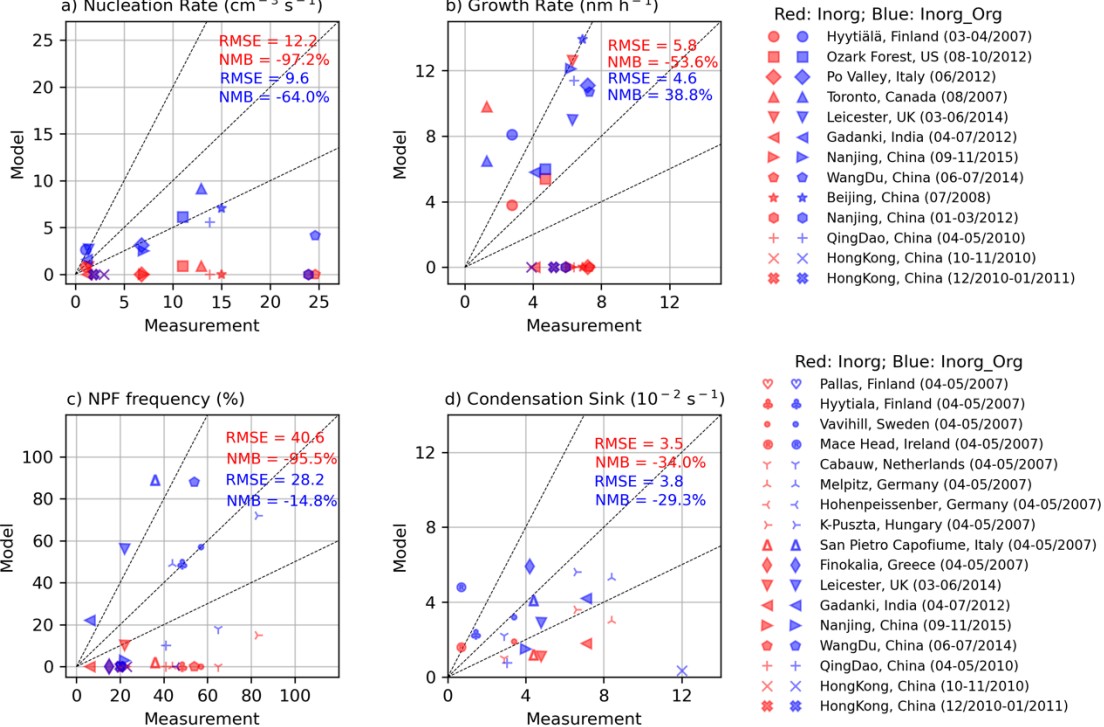

Figure 1: Comparison of simulated (choosing median value of measured smallest nucleation rates as threshold, see Text S4) and measured (a) nucleation rate, (b) growth rate, (c) NPF frequency, and (d) condensation sink in Inorg_Org (blue symbols) and Inorg (red symbols). Information regarding the measurement sites is summarized in Tables S1. Root mean square error (RMSE) and normalized mean bias (NMB) values are shown. (a) and (b) use the same dataset. (c) and (d) use the same dataset.

NPF events are far more frequent in Inorg_Org than in Inorg (NMB in NPF frequency changes from -96% to -15%) (Fig. 1c). Simulated NPF frequencies in Inorg_Org agree better with measurements in Europe (Hyytiälä, Vavihill, Cabauw, Melpitz, and K-Puszta) – see Table S5. Inorg_Org tends to overestimate NPF frequencies (by a factor of two) in some rural and forested

areas such as the San Pietro Capofiume, Leicester, and Wangdu and significantly underestimates frequencies in Chinese urban areas like Nanjing and Hong Kong (more than three times, Fig. 1a). These discrepancies are consistent with the nucleation rates discussed earlier (Table S4). In some locations (Melpitz, San Pietro Capofiume, Leicester and Gadanki) the overestimation of NPF frequency in Inorg_Org is consistent with an underestimation of CS and, vice versa, the underestimation of NPF frequency in Finokalia is consistent with an overestimation of CS (Figs. 1c and 1d).

## 3.2 Evaluation of aerosol and CCN number concentrations

To better understand the influence of implementing the new nucleation schemes, model-simulated particle number concentrations are evaluated against shipborne measurements over the ocean and CCN concentrations are evaluated using measurements from Amazonia. We also discuss the influence of uncertainties in HOMs chemistry. Unlike Spracklen et al. (2010), the lack of a nucleation mode in CAM means that we cannot use extensive measurements from condensation particle counters.

The values of number concentrations for particles with diameters larger than 10 nm (N10) in Inorg_Org agree better with measurements over the ocean (Figs. 2 and 3) but number concentrations for particles with diameters larger than 20 nm (N20) are overestimated in continental regions at the surface level (Fig. 4). From Figs. 2 and 3, N10 in Inorg_Org are the closest to measurements in both the North Atlantic and the Arctic when considering both normalized mean bias (NMB) and root mean square error (RMSE). Inorg_Org (Figs. 2e, f and 3e, f) alleviates both overestimation of N10 in Default resulting from their high sensitivity to $H_2SO_4$ concentrations (see Figs. 2a, b and 3a, b), and underestimation at some sites in Inorg caused by the lack of organics participating NPF (Fig. 2c, d and 3c, d). The influence of HOMs chemistry to N10 will be discussed in Section 3.3. In boreal Canada (Fig. 4), N20 are overestimated in Inorg_Org during the summer when NPF is particularly active. The overestimation is more significant at Egbert, located just 80 km north of Toronto, where Inorg shows better agreements with measurements. This discrepancy is likely due to an about 30% overestimation of growth rates (in Toronto, Table S4) and the underestimation of N50 during Northern hemisphere summer (as relevant to CS) at many stations in Inorg_Org (N50 in Fig. S9). The underestimation of N50 also explains the overestimation of N20 observed at the surface level in Amazonia in September 2014 (Fig. 4c) where insufficient large aerosols (N50, Fig. S9) result in a low CS, which in turn leads to an excessive number of aerosols merging into the Aitken mode (N30, Fig. S8).

After incorporating organic-related processes, CCN at high altitudes of Amazonia perform better compared with measurements. While all simulations considerably overestimate CCN concentrations at 0.5% supersaturation (0.5% ss) at surface level, CCN in the upper troposphere (5-12 km) in Inorg_Org (blue line, Fig. 4d) are the closest to measurements compared with both Default and Inorg. The more than 100% increase in CCN numbers at 8-12 km is mainly attributed to effective vertical transport of accretion products (ACC), which have a longer atmospheric lifetime compared to HOMs (Xu et al., 2022) and then participate in $J_{Org,n}$ and $J_{Org,i}$ (Figs. 6c and 6e), thereby increasing N20 and CCN concentrations at high levels. N20 shows the highest value at about 6 km altitude since nucleating species are abundant there and the condensation sink is relatively low (see CCN profile in Fig. 4d) compared to higher altitudes. Since we did not consider the suppression of C15 generated from isoprene and monoterpene-derived $RO_2$ (MT-$RO_2$) radicals cross-reactions on nucleation rates (Heinritzi et al., 2020), the ion-induced pure organic nucleation rate is overestimated in Amazon, and hence, cloud condensation nuclei (CCN) at surface level are overestimated in Inorg_Org.

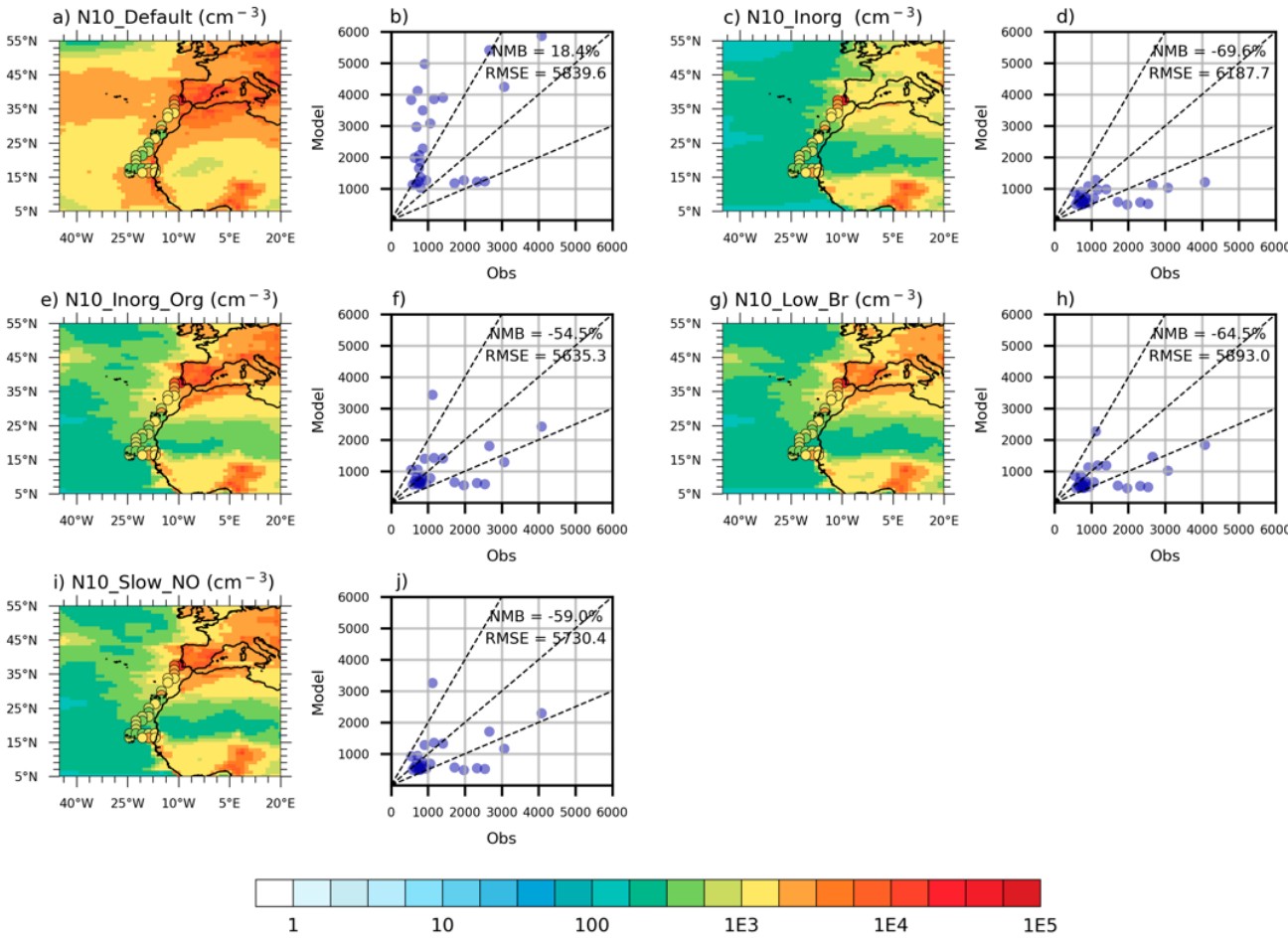

**Figure 2: N10 concentration in the North Atlantic from (a, b) Default, (c, d) Inorg, (e, f) Inorg_Org, (g, h) Low_Br, and (i, j) Slow_NO**
**(Unit: cm⁻³). N10 from RHaMBLe measurements are represented by filled circles. Model experiments are described in Table 2 and**
**model data come from mean value of May 2007 outputs. Numbers at the upper right (Fig. 2b, d, f, h and j) indicate normalized mean**
**bias (NMB) and root mean square error (RMSE).**

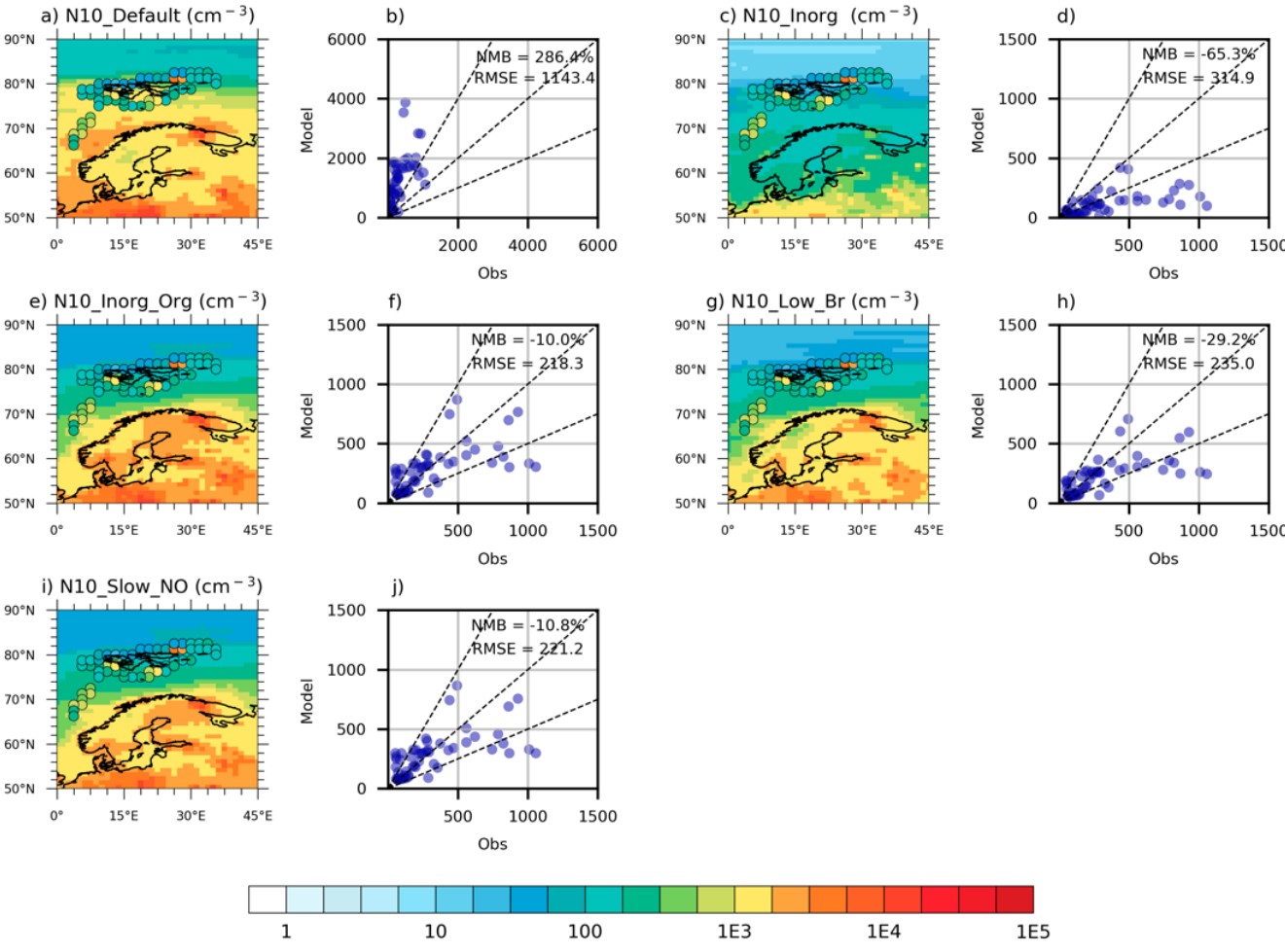

**Figure 3: N10 concentration in the Arctic from (a, b) Default, (c, d) Inorg, (e, f) Inorg_Org, (g, h) Low_Br and (i, j) Slow_NO (Unit: cm⁻³). N10 from ACCACIA measurements are represented by filled circles. Model experiments are described in Table 2 and model data come from mean value of July 2013 outputs. Numbers at the upper right (Fig. 3b, d, f, h and j) indicate NMB and RMSE.**

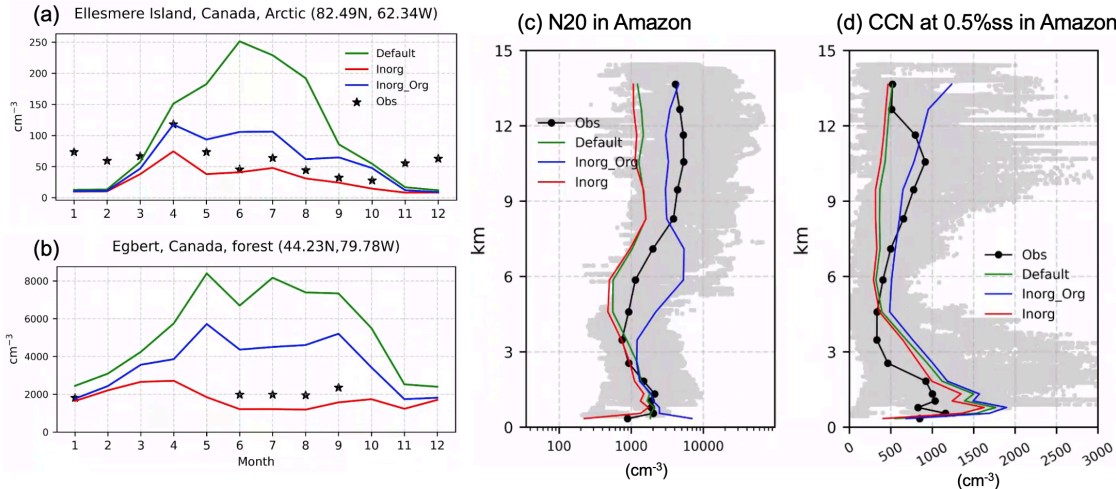

**Figure 4: Seasonal variation in 2013 at two Canadian sites (a) Ellesmere Island, and (b) Egbert. Vertical profiles of (c) N20 and (d) CCN at a supersaturation of 0.5% in the Amazon basin, measured at standard temperature and pressure (STP) (unit: cm⁻³), in September 2014.**

## 4 Quantifying the effect of organics-NPF on global aerosol

In this section, we use simulation results to quantify how HOMs affect the nucleation rate (Figs. 5 and 6), growth rate (Fig. 7), particle number (Fig. 8), and CCN number (Fig. 10) on a global scale. Additionally, results from sensitivity tests (Table 2) are also analyzed to reveal the influence of uncertainty from chemical mechanisms of HOMs and the relative importance of organic nucleation and initial growth process to particle number (Fig. 9).

Globally, the vertically-integrated (below 15 km) annual mean organic nucleation rate ($J_{Org,n} + J_{Org,i} + J_{SA-Org}$) in Inorg_Org is $32 \times 10^6$ cm⁻² s⁻¹ (Fig. 5a), closely matching the inorganic nucleation rate of $39 \times 10^6$ cm⁻² s⁻¹ (Table 4). $J_{SA-Org}$ contributes most to the total nucleation rate (45%), with an average value of $31.8 \times 10^6$ cm⁻² s⁻¹ (Table 4), and its spatial distribution (Fig. 6a) is influenced by both $H_2SO_4$ and HOMs concentrations. In regions abundant in HOMs (like boreal forests, North America and Australia in Fig. S5), the rate surges to $10^8$ cm⁻² s⁻¹ (Fig. 5a) and the organic contribution exceeds 80% (Fig. 5c). High concentrations of ACC are simulated in Amazonia (Fig. S6), where ACC are transported to high altitudes through strong convection (Section 3), thereby resulting in high rates of $J_{Org,i}$ (over 40%, Fig. 6f). $H_2SO_4$-$NH_3$ neutral nucleation comprises the largest proportion of the inorganic nucleation rate (>80% of inorganic and 40.5% of total nucleation rate, Table 4), particularly in China and India due to high anthropogenic $SO_2$ emissions. This is also consistent with the spatial distribution of $H_2SO_4$ (Fig. S4). The contribution of $H_2SO_4$-$NH_3$ neutral nucleation makes up more than 50% of the nucleation rate over coastal regions where HOMs and accretion products (ACC) are less abundant (Fig. S5 and S6) and the large proportion in Africa is due to high $NH_3$ column concentration (Luo et al., 2022).

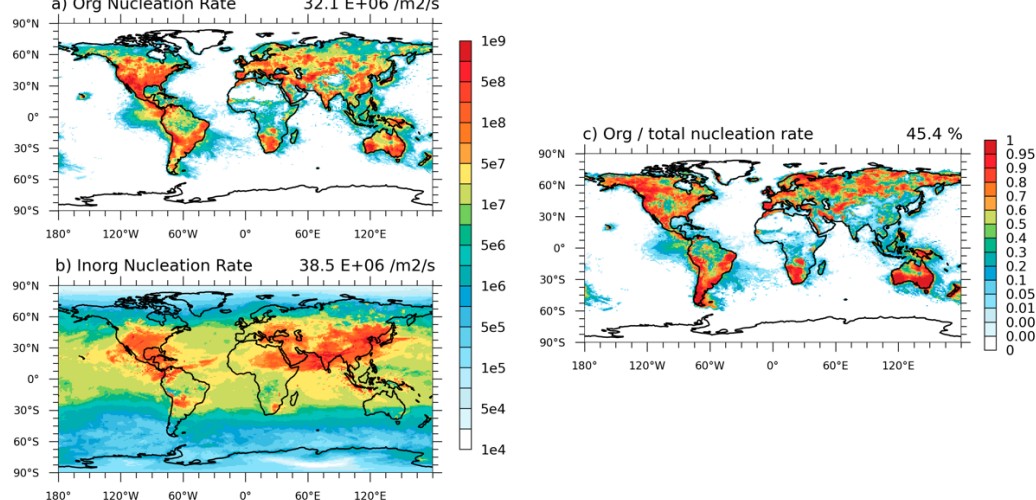

**Figure 5:** Spatial distribution of the 2013 annual mean nucleation rate ($j_{1nm}$, vertically-integrated below 15 km) attributed to (a) organics and (b) $H_2SO_4$. (unit: $m^{-2}$ $s^{-1}$) (c) is the proportion of organic nucleation proportion. Global mean values are shown on the top right of each figure.

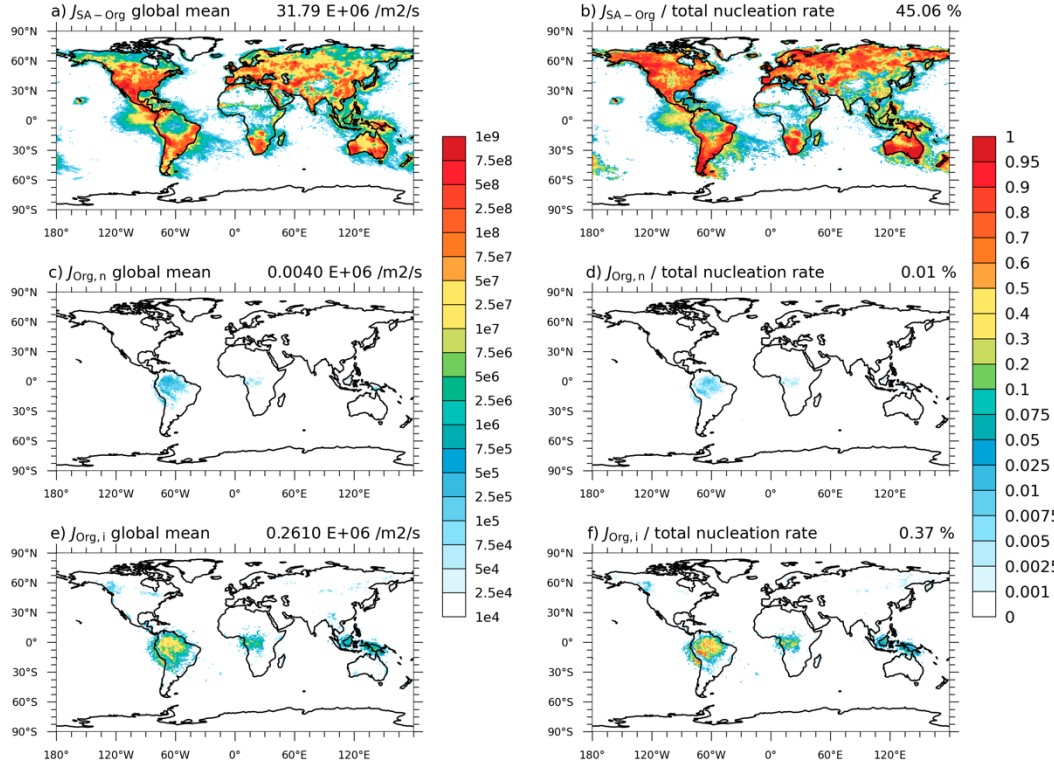

**Figure 6:** The 2013 annual average vertically-integrated organic nucleation rate ($j_{1nm}$) within the troposphere (a, c, e) (unit: $m^{-2}$ $s^{-1}$) and their respective contributions (b, d, f) for $J_{SA-Org}$ (a, b), $J_{Org,n}$ (c, d), and $J_{Org,i}$ (e, f) in the Inorg_Org. Global mean values are shown on the top right of each figure.

**Table 4. The 2013 annual average vertically-integrated organic nucleation rate ($j_{1nm}$) within the troposphere and its contributions to total nucleation rates in the Inorg_Org.**

| Pathways | Nucleation Rate (Unit: $10^6$ m$^{-2}\cdot$s$^{-1}$) | Proportion |
|---|---|---|
| $J_{SA}$ | 3.09 | 4.38% |
| $J_{SA,i}$ | 2.62 | 3.71% |
| $J_{SA-NH3}$ | 3.09 | 40.48% |
| $J_{SA-NH3,i}$ | 4.22 | 5.99% |
| $J_{Org,n}$ | 0.40 | 0.01% |
| $J_{Org,i}$ | 0.30 | 0.37% |
| $J_{SA-Org}$ | 31.80 | 45.06% |

Globally, the vertically-averaged (below 15 km) annual mean organic growth rate is 0.0048 nm h$^{-1}$ (summation of ACC and HOMs contribution). The organic growth rate contributes to 25% of the total growth rate for sub-20 nm particles. In regions such as Canada, the boreal forests, Amazonia, and Australia, where biogenic volatile organic compounds (BVOC) emissions dominate, organic growth accounts for over 60% of the total rate (Fig. 7c), consistent with the spatial distribution of HOMs and ACC (Figs. S5 and S6). Conversely, in China and India, H$_2$SO$_4$ exerts a predominant influence (> 90%) on the initial growth of new particles, with a rate of approximately 0.1 nm h$^{-1}$.

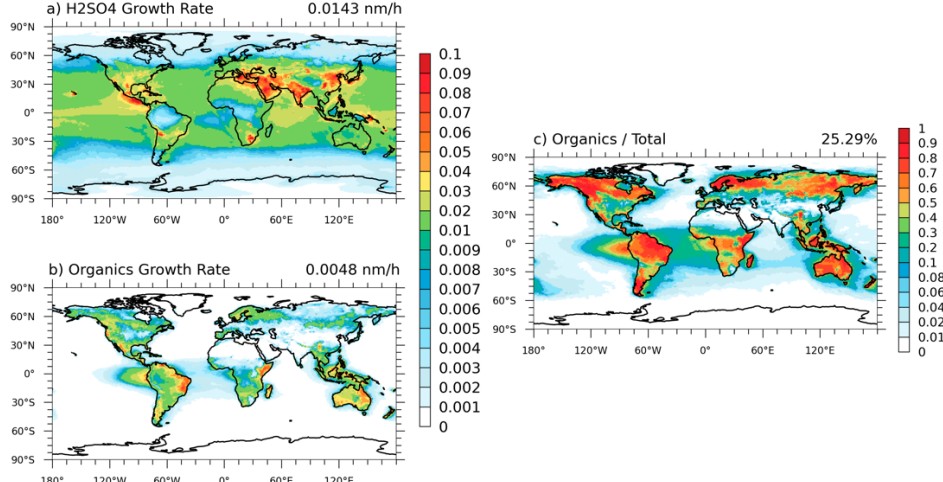

**Figure 7: Spatial distribution of the 2013 annual mean vertically averaged growth rate attributed to (a) H$_2$SO$_4$, and (b) organics, along with the percentage of organics contributions (c). Global mean values are shown on the top right of each figure.**

The global mean aerosol number burden increases by 39% (Fig. 8) in Inorg_Org compared to Inorg. The enhancement reaches

a maximum of 60% in Amazonia due to high $J_{Org,i}$ (Fig. 6f) driven by high ACC concentrations (Fig. S6). Results from the Low_Br and Slow_NO simulations reveal that the uncertainties in HOMs chemistry have a negligible effect on the total aerosol number concentrations. Relative to Inorg_Org, Low_Br leads to a 12% reduction in number concentrations (Fig. 9f), although the branching ratio of MT-RO$_2$ shifts significantly (from 80% to 25% for the MT+O$_3$ reaction and from 97% to 92% for the MT+OH reaction). The impact of slowing down the reaction rate of MT-HOM-RO$_2$ +NO (Slow_NO) is negligible globally (<

1%, Fig. 9h).

To evaluate the relative importance of organic contributions to nucleation and growth, we compare the Only_NR (no organic contribution to sub-20 nm particle growth rate) and Only_GR (no organic contribution to 1 nm particle nucleation rate) simulations. The global mean relative difference of aerosol number concentration between Only_GR (Fig. 9d) and Inorg_Org is

28% in a one-month simulation. Switching off growth alone (Only_NR) results in a 34% decrease in aerosol number relative to Inorg_Org (Fig. 9b). This illustrates that organic initial growth of new particles (sub-20 nm) is slightly more important than organic nucleation for the production of particles larger than 20 nm diameter.

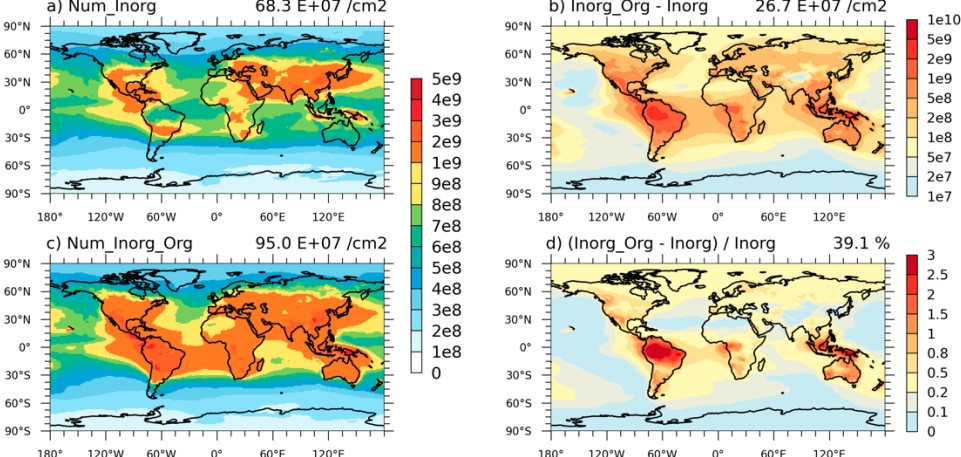

**Figure 8: Spatial distribution of annual mean total vertically-integrated particle number concentrations from (a) Inorg and (c) Inorg_Org (unit: cm$^{-2}$). Also, (b) change and (d) relative change are shown. Global mean values are shown on the top right of each figure.**

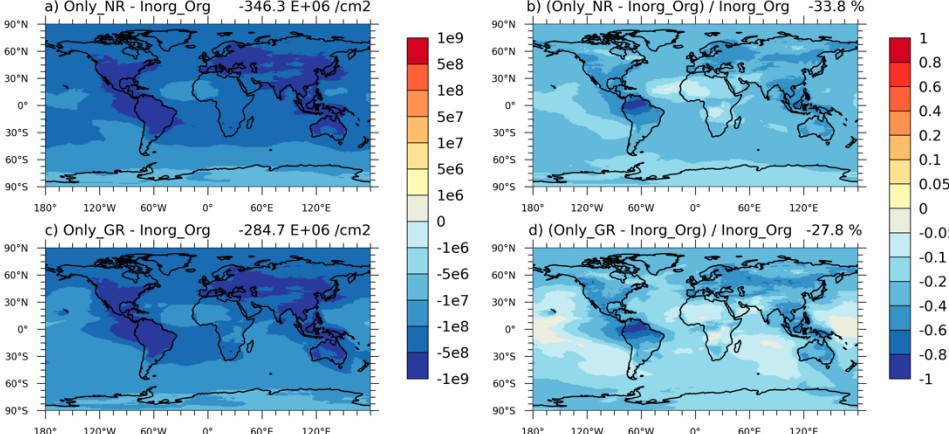

**Figure 9: Absolute differences (Units: cm$^{-2}$) and relative differences (Units: unitless) of in total vertically-integrated aerosol numbers in July 2013 between Inorg_Org and other sensitivity tests. Global mean values are shown on the top right of each figure. Model experiments are described in Table 2.**

The global annual average CCN burden at 0.5% supersaturation increases by 33% after adding organic NPF (Fig. 10). The spatial pattern of changes in CCN concentrations compared to Inorg is consistent with changes in aerosol number concentrations (Fig. 8), with increases predominantly occurring in regions abundant in HOMs and ACC (Fig. S5 and S6). Amazonia is the region most sensitive to organic-related processes due to high ACC concentrations (Fig. S6), where the total burden increases by more than 100% (Fig. 10d).

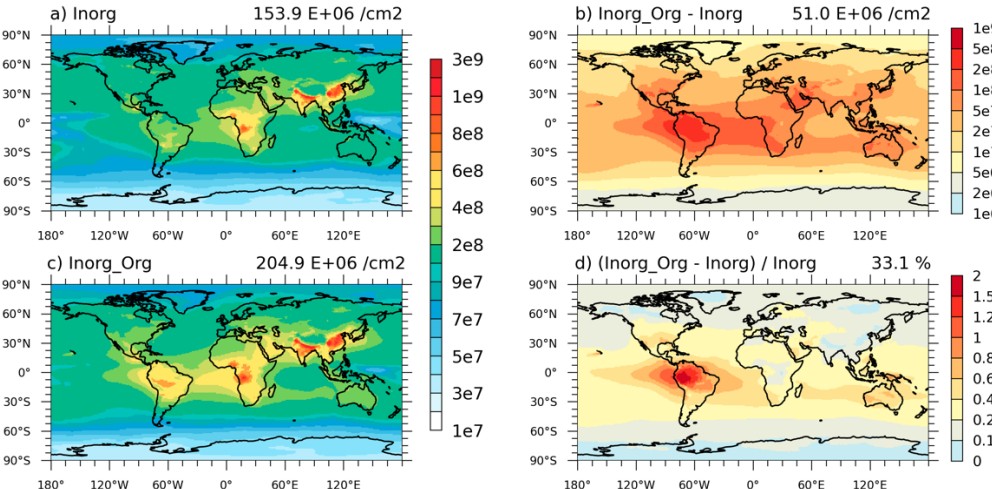

**Figure 10: Spatial distribution of annual mean total vertically-integrated CCN concentrations at 0.5% supersaturation for (a) Inorg and (c) Inorg_Org (unit: cm$^{-2}$). Also, (b) change and (d) relative change are shown. Global mean values are shown on the top right of each figure.**

## 5 Uncertainties from HOMs chemistry

This section aims to test the effects of using different autoxidation and self-/cross reaction rates as well as branching ratios during HOMs and ACC formation on the 1 nm nucleation rate, sub-20 nm growth rate, total aerosol number concentration and CCN number concentration.

The change of the autoxidation rate (Fast_auto and Slow_auto) affects both the nucleation and growth rates, particularly within HOMs source regions. A higher autoxidation rate leads to higher intermediate radical concentrations and thus more HOMs. Multiplying the autoxidation rate by 10 (Fast_auto) leads to a 6% increase in the nucleation rate and a 3% increase in the sub-20 nm growth rate on a global average (Fig. S11). The largest increases occur in regions such as the Amazon, Australia and boreal forests (>10%) where HOMs are most abundant. In these regions, the aerosol number concentration increased by more than 30% compared to the baseline Inorg_Org. Conversely, in Slow_auto, the nucleation and growth rates decline by 17% and 5%, respectively, resulting in a 15% reduction in aerosol number and a 6% reduction in CCN number globally. In the Amazon and boreal forests, these reductions exceed 20% (Table 5).

Adjusting the autoxidation temperature dependence to upper and lower limits (High_temp and Low_temp) causes changes in HOMs concentrations (Fig. S10). However, its impact on nucleation (-7% and -4% in High_temp and Low_temp), growth rate (-4% and ~0% in High_temp and Low_temp), aerosol (-12% and 3% in High_temp and Low_temp) and CCN (-4% and ~0% in High_temp and Low_temp) number concentrations is small (Fig. S11 and S12). This is because most of these changes occur over ocean, where $H_2SO_4$ has low concentrations. Consequently, the rate of heteromolecular nucleation of sulfuric acid and organics (HET), which is the greatest contributor to the organic-involving nucleation rate, does not show significant change in both experiments.

A lower dimerization reaction rate leads to decreased concentrations of accretion products (ACC), with a 71% decrease in Slow_accr. Lower consumption of MT-derived peroxyl radicals (MT-RO$_2$) during self-/cross- reaction means more of them can participate in autoxidation. This explains the higher HOMs concentration over source regions in Slow_accr compared to Inorg_Org (Fig. S10). However, the impact of slowing down the dimerization rate on aerosol and CCN number concentrations is negligible, remaining within 1% on a global average. In the Slow_accr experiment, the nucleation rate in the Amazon, where the concentration of ACC is highest, decreases by more than 50%, subsequently leading to a reduction of more than 20% in aerosol and CCN numbers (Table 5). This implies that both aerosol and CCN number concentrations in the Amazon basin are sensitive to the ACC concentration.

The change in HOMs and ACC concentrations in Slow_NO is almost negligible (-6% and 5%) hence its impact on aerosol and CCN number can be ignored (~0%). When the rate of NO termination is reduced, less MT-HOM-RO$_2$ is consumed when generating HOMs, and thus more MT-HOM-RO$_2$ participates in self-/cross-reactions. This explains the higher concentration of ACC and lower concentration of HOMs in Slow_NO compared to Inorg_Org. In contrast, in Low_Br, there is a significant

decrease in HOMs concentration (-51%) (Fig. S10) since the mass yield of MT-bRO$_2$ decreases, which subsequently leads to a 17% reduction in the nucleation rate (Fig. S11). Combining with approximately a 5% reduction in the growth rate of sub-20 nm particles, the concentrations of aerosol and CCN decrease by -12% and -5% respectively (Table 5). In regions most sensitive to biogenic HOMs chemistry, such as the Amazon, Australia, and boreal forests, the reduction in particle concentrations exceeds 20%.

**Table 5. Relative differences (Units: unitless) of vertically-integrated HOMs concentrations (HOMs), accretion products concentrations (ACC), nucleation (NR), growth rate (GR), aerosol number (Aerosol), and CCN number (CCN) in July 2013 between In-org_Org and other sensitivity tests. Values in the table are global mean values. Model experiments are described in Table 2.**

|           | HOMs | ACC  | NR   | GR   | Aerosol | CCN  |
|-----------|------|------|------|------|---------|------|
| Slow_NO   | -6%  | 5%   | -2%  | ~0   | ~0      | ~0   |
| Low_br    | -51% | -12% | -17% | -5%  | -12%    | -5%  |
| High_temp | -42% | -6%  | -7%  | -4%  | -12%    | -4%  |
| Low_temp  | 9%   | ~0   | -4%  | ~0   | 3%      | ~0   |
| Fast_auto | 75%  | 8%   | 6%   | 3%   | 18%     | 4%   |
| Slow_auto | -57% | -5%  | -17% | -5%  | -15%    | -6%  |
| Slow_accr | 66%  | -71% | -4%  | ~0   | -2%     | -1%  |

## 6 Comparison with previous studies

Our results show that vertically-integrated organic nucleation contributes 84% of the total nucleation rate within the lower 5.8 km of the atmosphere, which is much higher than that in previous studies (51% in Gordon et al. (2017) and 42% in Zhu and Penner (2019), Table 6). The HOMs concentrations in our simulations are about 10 times greater than in Gordon et al. (2017), as depicted in Fig. S7 and Fig. S2 in Gordon et al. (2016). Here we used a chemical mechanism of HOMs derived from chamber experiments (including both autoxidation and self-/cross-reactions of isoprene/monoterpene-derived radicals) while Gordon et al. (2016) estimated HOMs concentrations using an empirical fixed yield from monoterpene+O$_3$/OH. Higher HOMs concentrations in our simulation are much closer to measurements in Finland and the southeast USA (Fig. S4 and S5 in Liu et al. (2024)), and lead to higher $J_{\text{SA-Org}}$.

Updates to the inorganic nucleation scheme based on CLOUD chamber experiment data (Dunne et al., 2016) are the main reason we have higher contributions of vertically-integrated organic nucleation than Zhu and Penner (2019). The updated scheme decreases the inorganic nucleation rate by reducing its sensitivity to H$_2$SO$_4$ concentration. Thus, we simulate a higher organic nucleation proportion despite much lower $J_{\text{Org,i}}$ and $J_{\text{Org,n}}$ (Table 7). The lower values of $J_{\text{Org,i}}$ and $J_{\text{Org,n}}$ are caused by

our use of a more stringent definition of organic participation (only ACC due to their extreme/ultra-low volatility) in neutral and ion-induced pure organic nucleation (NON and ION, Eqs. (7) and (8)).

**Table 6. Fractions of NPF from organic and inorganic pathways are derived from Inorg_Org (annual average in 2013 below 5.8 km altitude). Results from Gordon et al. (2017) and Zhu and Penner (2019) are in present-day experiments.**

| Pathways | Below 5.8 km vertical integration | | |
|---|---|---|---|
| | Gordon et al. (2017) | Zhu and Penner (2019) | This study |
| $J_{SA}$ | ~0 | 58.40% | 0.03 % |
| $J_{SA,i}$ | 7.50 % | w/o[a] | 1.96 % |
| $J_{SA-NH3}$ | 17.00 % | w/o | 6.18 % |
| $J_{SA-NH3,i}$ | 24.00 % | w/o | 8.28 % |
| $J_{Org,n}$ | ~0 | 0.60% | ~0 |
| $J_{Org,i}$ | 4.10 % | 23.20% | 0.11 % |
| $J_{SA-Org,i}$[b] | 14.00 % | w/o | w/o |
| $J_{SA-Org}$ | 33.00 % | 17.80% | 83.44 % |

[a] w/o represents that there is no consideration of that nucleation scheme in publications

[b] $J_{SA-Org,i}$ represents that ion-induced heteromolecular nucleation of sulfuric acid and organics (HET).

**Table 7. Annually averaged NPF from three organic pathways: vertically-integrated results across the whole atmosphere.**

| Pathways | Zhu and Penner (2019)[a] | This study [b] |
|---|---|---|
| $J_{SA-Org}$ | 34.4 | 33.0 |
| $J_{Org,n}$ | 1.0 | 4.5 E-03 |
| $J_{Org,i}$ | 52.9 | 0.3 |
| Total | 88.2 | 33.2 |

[a,b] Results are compared between present-day atmospheres (Zhu and Penner, 2019) and 2013 annual mean in this study.

## 7 Summary and discussion

This study updates the inorganic nucleation scheme in CAM6-Chem according to chamber experimental measurements and
510 adds organic nucleation and initial growth scheme based on a state-of-the-art chemical mechanism for biogenic highly oxygenated molecules (HOMs) including autoxidation and self-/cross-reactions of isoprene/monoterpene-derived radicals. The organic nucleation scheme includes heteromolecular nucleation of sulfuric acid and organics (HET), neutral pure organic nucleation (NON), and ion-induced pure organic nucleation (ION). Organic condensation on sub-20 nm particles is also taken into account. The model was evaluated against new particle formation (NPF) events (occurrence frequency and nucleation and

515 growth rates) as well as aerosol and cloud condensation nuclei (CCN) number concentrations. Finally, we quantified the contribution of organics to nucleation rate, growth rate, aerosol and CCN number at 0.5% supersaturation globally.

Compared to the model with updated inorganic nucleation mechanisms (Inorg), the revised model with HOMs chemistry (Inorg_Org) agrees better with measurements of the nucleation rate and sub-20 nm particle growth rates at numerous sites
globally (the normalized mean bias (NMB) of nucleation rate changes from -97% to -64% and the NMB of growth rate changes from -96% to -15%, Fig. 1). Inorg_Org also simulates NPF event frequency in better agreements with measurements at 17 sites compared to Inorg (NMB changes from -96% to -15%, Fig. 1), thereby accurately reproducing N10 (number concentrations for particles with diameters larger than 10 nm) ship-borne measurements over the Arctic and North Atlantic (Figs. 2 and 3). Both N20 (number concentrations for particles with diameters larger than 20 nm) and CCN concentration increase more
than 100% between 8-12 km altitude (Fig. 4) over Amazonia after incorporating organic-related process and show better performance compared to aircraft measurements due to organic nucleating species (accretion productions) convection lifting to high level and then amplifying $J_{Org,i}$ (ION rate).

On a global scale, organics contribute 45% to the annual average vertically-integrated nucleation rate and 25% to the vertically
averaged initial growth rate from Inorg_Org (global mean). Compared to Inorg, Inorg_Org increases the annual average vertically-integrated aerosol number concentration by 39%. The simulation shows that the organic-related growth process exerts a more substantial influence on aerosol number than nucleation. These newly-formed particles result in a 33% increase in annual average vertically-integrated CCN concentrations at 0.5% supersaturation compared to Inorg. Both aerosol and CCN concentrations display the most significant increase in Amazonia, exceeding 60% and 100%, respectively, attributable to its
low aerosol concentration in Inorg in the background rainforest. More CCN produced through natural processes implies higher background aerosol abundance, thus weaker (less negative) historical aerosol forcing (Carslaw et al., 2013).

We also test the sensitivity of aerosol number concentrations to uncertainties from HOMs chemistry. Results show that including organic NPF processes in our model is more important than tuning these aspects of the parameterizations during HOMs
formation. Compared to the baseline Inorg_Org model, decreasing the branching ratio of the first-generation product from Monoterpene + $O_3$/OH, which could further undergo autoxidation (Low_Br), leads to only a 12% reduction in global average vertically-integrated aerosol number concentrations. Slowing down NO-involved chemical reactions due to NO concentration overestimation at two stations (Slow_NO) has very little effect on the global average aerosol number concentration (within ~1%) (Fig. 10). When altering the temperature dependence of autoxidation rate to a higher or lower value (High_temp and
Low_temp), HOMs concentrations change a lot (-42% and 9% respectively) but aerosol number concentrations only change a small amount (-12% and 3%). Factor of 10 changes of autoxidation rate (multiplying the autoxidation rate by 10 in Fast_auto and 0.1 in Slow_auto) results in relatively significant changes in the simulated aerosol number concentration (18% and -15% in global mean). When adjusting the dimerization rate coefficient of ACC formation to a lower value (Slow_accr), the aerosol

number change is negligible (within 2% on global average). Except for Amazon, the aerosol number concentrations are highly sensitive to ACC concentration and decrease by more than 20%.

The contribution of organic-involved nucleation to the vertically-integrated rate within the lower 5.8 km in our work (~83%) is significantly higher than previous studies. Compared to Gordon et al. (2017) (~51%), we use a more advanced HOMs chemistry that simulates higher HOMs concentrations in closer to measurements, thereby presenting higher $J_{SA-Org}$ (HET rate) and organics contribution. Compared to Zhu and Penner (2019) (~ 42%), we update the inorganic nucleation scheme based on CLOUD chamber experiments. Therefore, the inorganic nucleation rate and its proportion is reduced in our simulations and this provides a more reasonable baseline for the quantifying organic contribution. The greater contribution of biogenic organic nucleation to NPF implies that global aerosol may be more sensitive to changes in biogenic emissions. This finding should be tested with different representative concentration pathways (RCP) in the future, when human-induced global warming causes higher temperature and biogenic HOMs emissions, while emission reduction policies reduce anthropogenic emissions.

The chemical mechanisms of biogenic HOMs used in this study are state-of-the-art, but the largest uncertainties in this work still come from the chemical mechanisms of HOMs and thus HOMs concentration. Although we only consider two-step autoxidation reactions which are not the most advanced (Heinritzi et al., 2020; Simon et al., 2020), this impact on organic nucleation rate is almost negligible. Specifically, the number of autoxidation steps has almost no effect on the rate of heteromolecular nucleation of sulfuric acid and organics (HET), which is the most significant contributor to organic nucleation rate (Fig. 6 in the main text). This is mainly because the number of autoxidation steps affects neither the yield nor the concentration of C10-HOMs, only their molecular formulas and volatility. In our simulation, the lower volatility of C10-HOMs does not affect their participation in HET (i.e. LVOC, ELVOC and ULVOC can all contribute to HET), so the rate of HET is not influenced by the number of autoxidation steps. Previous studies (Kurtén et al., 2016; Tröstl et al., 2016) have already indicated that C10 class molecules alone do not have low enough vapor pressure to initiate the nucleation, without the presence of other species such as sulfuric acid or bases. This is further supported by the fact that C20 class molecules are mainly responsible for pure biogenic nucleation (Heinritzi et al., 2020; Frege et al., 2018). This means that C10-HOMs might become less volatile when undergoing one additional autoxidation step, transitioning from LVOC ($3 \times 10^{-5} < C^*(T) < 0.3$ µg m$^{-3}$, where $C^*(T)$ is the effective saturation concentration) to ELVOC ($3 \times 10^{-9} < C^*(T) < 3 \times 10^{-5}$ µg m$^{-3}$), but this is unlikely to affect the pure organic nucleation rate.

There might be some overestimations with C15 and C20 involved in new particle formation if we assume that all the accretion products are ELVOC or ULVOC. In the updated model, $C_{15}H_{18}O_9$ (C15, extremely low volatility) and $C_{20}H_{32}O_8$ (C20, ultra-low volatility) are just simplified representatives of all C15 and C20 dimers. While more dimer species with low volatility have already been detected on chamber experiments (Stolzenburg et al., 2018; Ye et al., 2019; Schervish and Donahue, 2020), they did not provide the explicit chemical kinetics of related reactions (i.e. the intermediate products and their yields) so it is

hard to incorporate these reactions and species in the model. On the other hand, although yields of accretion products vary by 1 to 2 orders of magnitude in previous studies (Rissanen et al., 2015; Berndt et al., 2018; Zhao et al., 2018), the yields of C15 and C20 currently use are very low (4%), resulting in relatively low dimer concentrations. Even if they were all ELVOC and ULVOC, it would not lead to a significant overestimation, and therefore, would not substantially impact nucleation and growth rates.

Neglecting the oligomerization and decomposition of accretion products may affect their concentrations; however, these effects are negligible. Not accounting for the oligomerization can lead to higher volatility of aerosols, resulting in a reduction of the mass concentration in the particle phase and condensation sink (CS), but increased mass in the gaseous phase. This could lead to an overestimation of the NPF rate. Since the mass of HOMs-SOA accounts for only about 10% of the total SOA mass, the impact on NPF rate can be neglected. Not considering decomposition may result in an overestimation of the mass and number concentration of HOMs in the particle phase, and consequently an overestimation of CS and an underestimation of the NPF rate. However, C15-SOA and C20-SOA account for less than 4% of the total SOA (Liu et al., 2024), this impact is also negligible.

We only implemented biogenic HOMs chemistry in the global model due to the limited knowledge of explicit chemical reactions forming anthropogenic-derived HOMs (Wang et al., 2017; Wang et al., 2020; Garmash et al., 2020; Molteni et al., 2018). This treatment likely leads to an underestimation of organic nucleation rates, particularly in urban areas (Fig. 1). More studies on chemical mechanisms of anthropogenic HOMs which could be applied in global model are needed. Our findings suggest that subsequent growth of the newly formed particles to larger sizes may have a more significant effect on aerosol number than nucleation. More studies are needed to quantify the contribution of anthropogenic organics to the initial growth rate. Changes in simulated aerosol number and size distribution caused by anthropogenic HOMs-driven NPF can have important implications for CCN concentrations and aerosol indirect forcing (Wang and Penner, 2009; Wang et al., 2009; Gordon et al., 2016; Zhu et al., 2019), which also need further analysis.

Besides organic species, the concentration of inorganic nucleating species will also affect the accuracy of the total nucleation rate. The overestimation of $H_2SO_4$ in CAM6-Chem could potentially impact our final results regarding the organic proportion in both nucleation and the initial growth rate because both the dependencies of inorganic and organic nucleation rate on $H_2SO_4$ concentration are modelled with an exponent greater than 2 (Eq. (2)-(6)). Also, ammonia ($NH_3$) emissions used in this study are adapted from the Community Emissions Data System (CEDS), which remain challenging to represent in models due to uncertainties, particularly in specific sectors. $NH_3$ emissions from human waste were adapted using methodologies from the Regional Emissions Inventory in Asia (REAS) (Kurokawa et al., 2013) and rely on a single global default emission factor. Not only is this emission factor uncertain, but there will certainly be regional variations due to differing environmental conditions that we were unable to take into account (Hoesly et al., 2018). For agricultural emissions, the actual practices of managing

livestock manure will affect true emissions; such practices vary significantly across the world but are not always well under-stood or reflected in the emission factors used in global inventories (Paulot et al., 2014). The aforementioned uncertainties in $NH_3$ will affect the inorganic nucleation rate and, consequently, the contribution of organics to the total nucleation rate.

In addition to the concentration of nucleating species, the uncertainty associated with NPF parameterization is also present in the model. Zhang et al. (2011) showed that radon contributes additional ionization in the boundary layer, especially over land. This implies that our pure organic nucleation rate might be underestimated since we only consider ion-pair production rate caused by galactic cosmic rays. This effect is negligible even over the continents since the contribution of ionization rate

caused by the radioactive decay of radon is only significant (> 30%) within the lowest 1 km (Fig. 12 in Zhang et al. (2011)). Above 3 km, the contribution of radon decay induced ionization rate can be neglected (<10%). In this study, we focused on the proportion of organic NPF in the vertical integration within the whole atmosphere. Therefore, we will not consider incor-porating the ion nucleation rate caused by radon.

The NPF rate at around 20 nm is calculated based on Eq. (14) from Kerminen and Kulmala (2002). This calculation is derived using several simplifying assumptions and approximations: (1) the only important sink for the newly formed particles is their coagulation with larger pre-existing particles; (2) the newly formed particles grow by condensation at a constant rate; (3) the pre-existing population of larger particles remains unchanged during the growth of the newly formed particles. However, Lehtinen et al. (2007) reformulated the previously published theory (Kerminen and Kulmala, 2002) to better account for the

size dependence of the loss rates of newly formed particles (i.e., coagulation sink), rather than simplifying it as the gas con-densation sink (CS). The uncertainty range caused by using the constant CS to replace the size-dependent coagulation sink (CoagS) is shown in Text S5. Recent studies (Stolzenburg et al., 2020; Ozon et al., 2021; Deng et al., 2020) also show that aerosol growth rates are not constant with size. However, CAM6-Chem does not include a nucleation mode, which means that newly formed particles grow from 1.7 nm to 20 nm (geometric diameter in Aitken mode) within one physical timestep (30

minutes), making it impossible to resolve the growth rates of sub-20 nm particles. Future work, such as implementing a nucle-ation mode in CAM6-Chem and resolving particle growth rate within 20 nm, is therefore worth exploring.

**Data availability**

The data set from the ACRIDICON–CHUVA campaign is archived and publicly accessible from the HALO database maintained by the German Aerospace Center (DLR) at https://halo-db.pa.op.dlr.de/mission/5 (Full description of the data set is shown in Andreae et al. (2018)). Data processed into a consistent, model-ready format during the NERC-funded GASSP project (NE/J024252/1) are available upon request from co-authors Leighton Regayre and Ken Carslaw

## Author contribution

MW and XD designed the study. XS performed the data analysis, produced the figures, and wrote the manuscript draft. LR, MA, MP and MY collected the dataset. YL, WS, SA and KS contributed to the analysis methods. DJ provided the model. All the authors contributed to discussion, writing, and editing of the manuscript.

## Competing interests

At least one of the (co-)authors is a member of the editorial board of Atmospheric Chemistry and Physics. The contact author has declared that none of the authors has any competing interests.

## Acknowledgments

This research is supported by the Natural Science Foundation of China (41925023, U2342223, and 91744208), the Collaborative Innovation Center of Climate Change, Jiangsu Province, and the Fundamental Research Funds for the Central Universities - CEMAC "GeoX" Interdisciplinary Program (2024ZD05). Leighton Regayre was supported by the Met Office Hadley Centre Climate Programme funded by DSIT. We greatly appreciate the High Performance Computing Center of Nanjing University for providing the computational resources used in this work. The CESM project is supported primarily by the United States National Science Foundation (NSF). This material is based upon work supported by the National Center for Atmospheric Research, which is a major facility sponsored by the NSF under Cooperative Agreement No. 1852977. We thank all the scientists, software engineers, and administrators who contributed to the development of CESM2.

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
