# Peer review of "Global modeling of aerosol nucleation with a semi-explicit chemical mechanism for highly oxygenated organic molecules (HOMs)"

_EGUsphere, 2024_

## Author Comment (AC2)

We are very grateful to the evaluations from the reviewers, which have allowed us to clarify and improve the manuscript. Below we addressed the reviewer comments, with the reviewer comments in black and our response in blue.

Before we provide the detailed point by point reply, we provide an overview of the main changes and improvements:

1. We added some detailed descriptions of the model and model output in Section 2 Data and Method.

2. We added four sensitivity tests related to reaction rate during HOMs formation and their description in Section 2.4.

3. We added a new Section to discuss the impact of uncertainties from HOMs chemistry on aerosol and CCN number.

4. In the last Section (Summary), we added more discussion on the limitations and uncertainties associated with our current results.

**Reply for the Anonymous referee #2**

**General Comments:**

A version of the CESM climate model with state-of-the-art new particle formation (NPF) mechanisms is presented, with a focus on the production of Highly Oxygenated organic Molecules (HOMs). The model demonstrates improved agreement with observations. The authors find that organic molecules play a more important role in global NPF than previous studies suggested. Table 5 suggests 83.44% of nucleation proceeds via the mixed $H_2SO_4$-organic pathway below 5.8km, a result that, if nothing else, highlights the importance of further studying this possible NPF pathway.

While not emphasized in the paper, the authors also include an upgraded inorganic NPF mechanism, a potentially very useful innovation.

The article documents a significant effort and it is novel for this level of complexity in new particle formation to be included in a global climate model. The analysis and model evaluation are of high quality with some useful innovations such as the NPF event threshold.

I recommend the paper for publication, subject to responses to the comments below. I also appreciate that, while I do suggest some more sensitivity studies, it is surely not within the scope of the paper to explore all possible uncertainties, as long as the limited nature of the sensitivity studies is properly discussed.

**Response:** We would like to thank the referee for providing the insightful suggestions, which indeed help us further improve the manuscript. We have added required discussion to account for the major and minor comments and marked the corresponding line number in the revised paper. Please see the revision and the response for the comments as follow.

**Major Comment#1**: What are the uncertainties associated with using the Kerminen and Kulmala (2002) approximation for aerosol growth rates up to 20nm in diameter? Many models, including some GCMs that participated in CMIP6, resolve aerosols prognostically and advect them at much smaller sizes. It seems that using the Kerminen and Kulmala (2002) formula in this way for precise studies of NPF might incur quite large biases. Growth rates are not constant with size (Stolzenburg et al, ACP 2020, e.g. Eq. 16), and the more accurate version of the equation that takes better account of the size dependence of the loss rates is given by Lehtinen et al (J. Aerosol Science 2007). I suspect reasonable bounds on these potentially large uncertainties could be determined with some sensitivity studies, without changing the structure of the model.

Also, (L173) should the 0.66 factor be modified between default and updated NPF mechanisms to account for the difference between the 1.7nm starting size for the updated scheme and the 1nm starting size for the default scheme? I believe the parameterization depends on $(1/1.7 – 1/20)$ or $(1/1 – 1/20)$ so this likely makes quite a big difference.

**Response:** We apologize for the typo in Eq. (9) and it is corrected to:

$$ j_{20\text{nm}} = j_{1.7 \text{ nm}} \exp\left[ -\left( \frac{1}{1.7} - \frac{1}{20} \right) \frac{\gamma \text{CS}}{\text{GR}} \right] \tag{9} $$

However, without changing the structure of the model, we cannot simulate growth rates with size dependence and take account of the size dependence of the loss rates. The main reason is that there is no nucleation mode in CAM6-Chem. For Aitken mode, the geometric diameter is about 20 nm so we cannot resolve aerosols growth rate within 20nm. We have added the assumption and limitation at the **Summary**:

"The NPF rate at about 20nm is calculated based on the Eq. (14) based on Kerminen and Kulmula (2002). It is derived with several simplifying assumptions and approximations:

(1) the only important sink for the newly formed particles is their coagulation to larger preexisting particles;

(2) the newly formed particles grow by condensation at a constant rate;

(3) the preexisting population of larger particles remains unchanged during the newly formed particles growth.

However, Lehtinen et al. (2007) reformulates the previously (Kerminen and Kulmala, 2002) published theory, in which both the accuracy has been improved and the applicability to different conditions is more straightforward. Since CAM6-Chem does not include a nucleation mode, we are not able to

update the NPF rate based on Lehtinen et al. (2007). However, future work, such as implementing a nucleation mode in CAM6-Chem and resolving particle growth rate within 20nm, is worth exploring."

**Reference**

Lehtinen, K. E. J., Dal Maso, M., Kulmala, M., and Kerminen, V.-M.: Estimating nucleation rates from apparent particle formation rates and vice versa: Revised formulation of the Kerminen–Kulmala equation, J. Aerosol Sci., 38, 988-994, https://doi.org/10.1016/j.jaerosci.2007.06.009, 2007.

**Major Comment#2**: The model evaluation has some missing details. How were number concentrations of 10nm particles calculated in the model for comparison with measurements? At what temporal frequency were variables written out of the model for calculating nucleation rates and aerosol number concentrations? What kind of interpolation was done to match model with observation stations and aircraft measurements?

**Response:** Thanks for suggestions. We added the following sentences in **Section 2.3**:

"CAM6-Chem utilizes a four-mode version of the Modal Aerosol Module (MAM4) (Liu et al., 2016), including Aitken mode (with diameter 9~52nm), accumulation mode (54~480 nm), coarse mode (400~40000 nm) and primary mode (10~100nm). The integral concentration from 0 to $r_p$ is computed using the error function (erf):

$$N_{>r_p} = N_{mode} \left( \frac{1}{2} + \frac{1}{2} erf \left( \frac{x}{\sqrt{2}} \right) \right) \tag{12}$$

where $x = \ln(r_p/r_m)/\ln\sigma$. $\sigma$ is the geometric standard deviation (the width) of the lognormal distribution $r_m$ is the median radius of the mode. The integral concentration above $r_p$ is therefore $N_{>r_p} = N_{mode} - N_{<r_p}$.

The temporal frequency of the nucleation rate, growth rate, and condensation sink written out of the model are hourly, and the time periods of the model simulation are consistent with the observation period (with an additional month for spin-up). For aerosol number concentrations (including over oceans and land), the model outputs data on a monthly basis, and we compare these monthly averages with observations. When comparing the aerosol and CCN number concentrations with the field campaign in the Amazon basin, the output frequency from model is hourly. Then, we slice the aircraft measurements of aerosol and CCN number concentrations vertical profiles according to the model

output dimensions (4 dimensions including time, height, latitude and longitude). We average all measurements data within each slice and compare it with the corresponding model output data."

**Reference**

Liu, X., Ma, P. L., Wang, H., Tilmes, S., Singh, B., Easter, R. C., Ghan, S. J., and Rasch, P. J.: Description and evaluation of a new four-mode version of the Modal Aerosol Module (MAM4) within version 5.3 of the Community Atmosphere Model, Geosci. Model Dev., 9, 505-522, 10.5194/gmd-9-505-2016, 2016.

**Major Comment#3**: At line 330, I think the results suggest most ACC does not undergo autoxidation even though it has 8 or 9 oxygen atoms (Table 1), because ACC are unaffected by the NO sensitivity study. That may be possible, though it seems rather surprising. Also I would imagine ACC which are autoxidized will usually have more oxygen han those which don't, so will be more likely to be ULVOC and nucleate. Either way it doesn't make sense to me that uncertainties in organic chemistry (ACC as well as HOM) do not affect CCN number in Amazonia where pure biogenic NPF dominates – it seems more likely that the authors were unable to sample the uncertainty space sufficiently, and this could be discussed more.

**Response:** The formation of and C15 and C20 dimer (accretion products, ACC) requires MT-aRO$_2$, MT-bRO$_2$, MT-cRO$_2$ and MT-HOM-RO$_2$ (MT-RO$_2$). Both MT-cRO$_2$ and MT-HOM-RO$_2$ are generated through autoxidation. During the autoxidation process, the number of oxygen atoms in these MT-RO$_2$ continuously increase, hence ACC contains 8-9 oxygen atoms.

In the NO sensitivity study, although there is no change in the rate of autoxidation reactions that generate MT-RO$_2$, the reaction rate of the NO termination pathway affects the concentration of MT-HOM-RO$_2$, and consequently, the concentration of ACC.

We appreciate the referee for pointing out this issue, which was also mentioned by another referee. In order to test the impact of the ACC chemistry on nucleation and growth rate as well as aerosol and CCN number concentration, we have run one additional sensitivity test based on the lower limits of self-/cross-reaction rates provided in chamber experiments (Weber et al., 2020; Berndt et al., 2018), labelled "Slow_accr". Please refer to the response or major comment#1 of referee #1 where more details are presented regarding the sensitivity tests.


Indeed, I would speculate despite the authors' findings, CCN formation in general could still be very sensitive to either the chemical formation of organic molecules (HOMs and ACC) or to the choices made by the modelers on which species participate in NPF. In particular, it is speculative that all HOMs (as defined by the authors) participate in NPF with $H_2SO_4$ at the rate specified in the Riccobono et al (2014) CLOUD study, as in Eq 6 of this paper. The species defined by Riccobono et al was called 'BioOxOrg', not HOM. The large uncertainties associated with these parameterizations should be acknowledged in this paper, and ideally quantified with sensitivity studies.

I would also speculate that the results are likely sensitive to uncertainties in concentrations of $H_2SO_4$ and $NH_3$. The sources of these uncertainties are not really discussed in the main text yet (despite documentation that a bias in $H_2SO_4$ exists and a helpful table of budgets in the supplement).

In the light of these remarks, while I do agree that the uncertainty due to the omission of anthropogenic organics is potentially important, it is far from the only important uncertainty, and the last paragraph of the paper could be more balanced.

**Response:** Yes, the species defined by Riccobono et al. (2014) was called 'BioOxOrg' instead of "HOMs" since at that time the measurement technique was not able to distinguish the nucleating organic species participating in different organic nucleation scheme. We are grateful to the referee for highlighting these important uncertainties and the discussion of both the chemical formation of organic molecules and the choice of nucleating species have already added in new section **"5 Uncertainties from HOMs chemistry"** and the **Summary**. Please refer to the response or major comment#1 of referee #1, #2, #3, #4 and #5 where more details are presented regarding these uncertainties.

Also, uncertainties in concentrations of $H_2SO_4$ and $NH_3$ have added in the **Summary**:

"The overestimation of $H_2SO_4$ in CAM6-Chem could potentially impact our final results regarding the organic proportion in both nucleation and the initial growth rate because both the dependency of inorganic and organic nucleation rate on $H_2SO_4$ concentration are modeled with an exponent greater than 2 (Eq. (2)-(6)). Also, nitrate is not included CMIP6 emissions because of uncertainties in both ammonia emissions and its chemistry and removal (Heald et al., 2012; Paulot et al., 2016). The underestimated nitrate concentrations result in reduced rate of ammonia consumption, potentially leading to an overestimation of residual atmospheric ammonia. Therefore, the inorganic nucleation rate may be overestimated and consequently, the organic proportion of the nucleation rate is likely underestimated."

**Minor Comment#6**: L320 misspelling of species

**Response:** We apologize for the spelling error. We have corrected this.

---

## Author Response (AR1)

We are very grateful to the evaluations from the reviewers, which have allowed us to clarify and improve the manuscript. Below we addressed the reviewer comments, with the reviewer comments in black and our response in blue.

Before we provide the detailed point by point reply, we provide an overview of the main changes and improvements:

1. We added some detailed descriptions of the model and model output in Section 2 Data and Method.

2. We added four sensitivity tests related to reaction rate during HOMs formation and their description in Section 2.4.

3. We added a new Section to discuss the impact of uncertainties from HOMs chemistry on aerosol and CCN number.

4. In the last Section (Summary), we added more discussion on the limitations and uncertainties associated with our current results.

**Reply for the referee comment#1**

**General comments:** This paper details work on implementing a HOM chemistry scheme into a large-scale model and comparing the results with an updated inorganic-only nucleation scheme. The inclusion of organic nucleation and growth at small particle sizes due to HOM formation leads to better agreement overall with measurements of high altitude CCN and frequency of NPF events globally. This work provides an interesting look into incorporating HOM chemistry into global models. The results seem reasonable and fit within the scope of ACP. My main concern stems from a lack sensitivity studies surrounding many of the uncertainties in the mechanism as well as a lack of discussion of the limitations of the mechanism. Given HOM chemistry is an active area of research, it makes these results incredibly significant to the community, but also means communicating clearly the limitations of the work given the present understanding of HOM chemistry is all the more important. Overall, I think this paper presents an important contribution to the field and I would support publication if the following comments are addressed.

**General Response:** We greatly appreciate the referee for their time and efforts devoted to the review of our submission. We realize that most of the comments are due to the missing sensitivity studies of exploring possible uncertainties from HOMs chemistry. We will present these details in the following responses.

**Specific comments and responses:**

**Major Comment#1:** The sensitivity studies done in this work seem informative, but more should be done on the other unknown parameters related to HOM formation such as the autoxidation rate coefficient, the temperature-dependence of this rate coefficient, and the dimerization rate coefficients.

**Response:** We agree with the reviewer the HOMs concentrations are affected by many parameters. We have added a discussion section (**Section 5**) about the potential uncertainties in autoxidation rate, autoxidation temperature dependence, and dimerization reaction rate with five extra sensitivity tests. The discussion of the original sensitivity tests related to HOMs chemistry (i.e. "Slow_NO" and "Low_Br") were also moved to this new Section (**Section 5**) as all of them are related to uncertainties from organic chemistry. Details of five additional sensitivity tests are listed in two new tables in the **Supplementary Information** (Table S7 and S8)

Table 2 in **Section 2.4 (Sensitivity experiments)** was also modified as follows (added five sets of sensitivity experiments, labeled "High_temp", "Low_temp", "Fast_auto", "Slow_auto" and "Slow_accr" in the last five lines):

**Table 1. Configurations of CESM2.1.0 Experiments**

| Test Name | Updated inorganic nucleation | HOMs chemistry | Organic Nucleation | Organic Growth | Other Changes |
|---|---|---|---|---|---|
| Default | × | × | × | × | / |
| Inorg | ✓ | ✓ | × | × | / |
| Inorg_Org | ✓ | ✓ | ✓ | ✓ | / |
| Only_NR | ✓ | ✓ | ✓ | × | / |
| Only_GR | ✓ | ✓ | × | ✓ | / |
| Low_Br | ✓ | ✓ | ✓ | ✓ | Lower branch ratio of the first-generation product (MT-RO$_2$) from MT + O$_3$ and MT + OH, which could be further auto-oxidized |
| Slow_NO | ✓ | ✓ | ✓ | ✓ | Rate of MT-HOM-RO$_2$ (second-generation autoxidation product) + NO generating HOMs, multiplied by 0.2 |
| High_temp | ✓ | ✓ | ✓ | ✓ | Autoxidation rate with high temperature dependence (Roldin et al., 2019) (Table S7) |
| Low_temp | ✓ | ✓ | ✓ | ✓ | Autoxidation rate with low temperature dependence (Weber et al., 2020) (Table S7) |
| Fast_auto | ✓ | ✓ | ✓ | ✓ | Autoxidation rate multiplied by 10 |
| Slow_auto | ✓ | ✓ | ✓ | ✓ | Autoxidation rate multiplied by 0.1 |

| | | | | | |
|---|---|---|---|---|---|
| Slow_accr | ✓ | ✓ | ✓ | ✓ | Using slower self-/cross reaction rate derived from Weber et al. (2020) and Berndt et al. (2018) (Table S8) |

The description of these sensitivity tests (Table 2) in **Section 2.4** were modified as (The underlined content is newly added or modified):

"We also conducted five sensitivity simulations to examine uncertainties in concentrations of HOMs (Table 2): sensitivity to the branching ratio from the first generation of monoterpene (MT) reactions with $O_3$/OH that can be auto-oxidized (Low_Br), sensitivity to the rate of termination reaction involving NO (Slow_NO), sensitivity to the autoxidation temperature dependence (High_temp and Low_temp), sensitivity to the autoxidation rate (Fast_auto and Slow_auto) and sensitivity to the self-/cross-reaction rate (Slow_accr) (Table 2). In Inorg_Org, the branching ratios for the MT-derived peroxyl radicals (MT-bRO$_2$) which could be further auto-oxidized are set at 80% for MT+O$_3$ and 97% for MT+OH reactions, corresponding to the high values reported in Xu et al. (2022). In the Low_Br simulation (Table 2), the branching ratio for MT-RO$_2$ is set as 25% for MT + O$_3$ and 92% and MT + OH. Both the high and low branching ratios fall within the range of previous studies (Lee et al., 2023; Pye et al., 2019; Weber et al., 2020; Xu et al., 2018; Jokinen et al., 2015; Roldin et al., 2019). In Slow_NO, the reaction rate of MT-HOM-RO$_2$+NO (MT-HOM-RO$_2$, the second-generation product of autoxidation (Text S1) is set as one-fifth of that in Inorg_Org, given that the simulated NO concentration is fourfold higher than the measured values in the boreal forest in Finland and in the southeast USA (Fig. S3 and S2 in Liu et al. (2024)). In High_temp and Low_temp, the temperature dependence of autoxidation rate are set to lower and upper limits (i.e. representing possible higher and lower bound of activation energy, Table S7) based on chamber experiments (Roldin et al., 2019; Weber et al., 2020). In Fast_auto and Slow_auto, the autoxidation reaction rates are multiplied by 10 and 0.1 respectively. In Slow_accr, the rate of self-/cross- reactions are set as the lower value (Table S8) based on chamber experiments (Weber et al., 2020; Berndt et al., 2018)."

We have added a new section "**5 Uncertainties from HOMs chemistry**". The content is as follows:

[revised manuscript text omitted]

Table S7 and S8 and Figure S10, S11 and S12 are shown as follow:

Table S7. Description of sensitivity test for the autoxidation rate with different temperature dependency (Roldin et al., 2019; Weber et al., 2020).

| Test Name | Reaction rate for generating | |
|---|---|---|
|  | MT-cRO$_2$ | MT-HOM-RO$_2$ |
|  | (first-generation autoxidation products) | (multi-generation autoxidation products) |
| Low_temp | 1.009E9*exp(-6000/T) | 9.500E8*exp(-6000/T) |
| High_temp | 7.768E17*exp(-12077/T) | 7.311E17*exp(-12077/T) |
| Inorg_Org | 9.800E12*exp(-8836/T) | 9.800E12*exp(-8836/T) |

Table S8. Description of sensitivity test for the different self-/cross- reaction rate (Weber et al., 2020; Berndt et al., 2018).

| Test Name | Reaction rate for generating | |
|---|---|---|
|  | C15 | C20 |
| Slow_accr | 1.800e-12 | 0.400e-11 |
| Inorg_Org | 2.000~4.000e-11 | 4.000~26.000e-11 |

We are grateful to the referee for highlighting these important uncertainties of HOMs and ACC chemistry so we have included the following discussion at the **Summary**:

"We also test the sensitivity of aerosol number concentrations to uncertainties from HOMs chemistry. Results show that including organic NPF processes in our model is more important than tuning these aspects of the parametrizations during HOMs formation. Compared to the baseline Inorg_Org model, decreasing the branching ratio of the first-generation product from Monoterpene+$O_3$/OH, which could further undergo autoxidation (Low_Br), leads to only a 12% reduction in global average vertically-integrated aerosol number concentrations. Slowing down NO-involved chemical reactions due to NO concentration overestimation at two stations (Slow_NO) has very little effect on the global average aerosol number concentration (within ~1%) (Fig. 10). When altering the temperature dependence of autoxidation rate into higher or lower value (High_temp and Low_temp), HOMs concentrations change a lot (-42% and 9% respectively) but aerosol number concentrations only change a small amount (-12% and 3%). Factor of 10 changes of autoxidation rate (multiplying the autoxidation rate by 10 in Fast_auto and 0.1 in Slow_auto) results in a relatively significant changes in the simulated aerosol number concentration (18% and -15% in global mean). When adjusting the dimerization rate coefficient of ACC formation to lower value (Slow_accr), the aerosol number change is negligible (within 2% on global average). Except for Amazon, the aerosol number concentrations are highly sensitive to ACC concentration and decrease by about more than 20%."

[Figure]

Figure S10. Relative differences (Units: unitless) of vertically-integrated HOMs (left column) and accretion products (right column) in July 2013 between Inorg_Org and other sensitivity tests. Global mean values are shown on the top right of each figure. Model experiments are described in Table 2.

[Figure]

Figure S11. Relative differences (Units: unitless) of vertically-integrated nucleation (left column) and growth rate (right column) in July 2013 between Inorg_Org and other sensitivity tests. Global mean values are shown on the top right of each figure. Model experiments are described in Table 2.

[Figure]

Figure S12. Relative differences (Units: unitless) of vertically-integrated aerosol (a, c, e, g, I and k) and CCN number concentration (b, d, f, h, j and l) in July 2013 between Inorg_Org and other sensitivity tests. Global mean values are shown on the top right of each figure. Model experiments are described in Table 2.

**Reference**

Berndt, T., Mentler, B., Scholz, W., Fischer, L., Herrmann, H., Kulmala, M., and Hansel, A.: Accretion Product Formation from Ozonolysis and OH Radical Reaction of α-Pinene: Mechanistic Insight and the Influence of Isoprene and Ethylene, Environ. Sci. Technol., 52, 11069-11077, 10.1021/acs.est.8b02210, 2018.

Xu, R. C., Thornton, J. A., Lee, B., Zhang, Y. X., Jaegle, L., Lopez-Hilfiker, F. D., Rantala, P., and Petaja, T.: Global simulations of monoterpene-derived peroxy radical fates and the distributions of highly oxygenated organic molecules (HOMs) and accretion products, Atmos. Chem. Phys., 22, 5477-5494, 10.5194/acp-22-5477-2022, 2022.

Jokinen, T., Berndt, T., Makkonen, R., Kerminen, V. M., Junninen, H., Paasonen, P., Stratmann, F., Herrmann, H., Guenther, A. B., Worsnop, D. R., Kulmala, M., Ehn, M., and Sipila, M.: Production of extremely low volatile organic compounds from biogenic emissions: Measured yields and atmospheric implications, P. Natl. Acad. Sci. USA, 112, 7123-7128, 10.1073/pnas.1423977112, 2015.

Lee, S., Shin, J. E., Yoon, R., Yoo, H., and Kim, S.: Annulation of O-silyl N,O-ketene acetals with alkynes for the synthesis of dihydropyridinones and its application in concise total synthesis of phenanthroindolizidine alkaloids, Front. Chem., 11, 1267422, 10.3389/fchem.2023.1267422, 2023.

Liu, Y., Dong, X., Wang, M., Xu, R., Thornton, J. A., Shao, X., Emmons, L. K., Jo, D. S., Yue, M., and Shrivastava, M.: A Modeling Study of Global Distribution and Formation Pathways of Highly Oxygenated Organic Molecules Derived Secondary Organic Aerosols (HOMs-SOA) from Monoterpenes, J. Geophys. Res.-Atmos., (under review), 2024.

Pye, H. O. T., D'Ambro, E. L., Lee, B. H., Schobesberger, S., Takeuchi, M., Zhao, Y., Lopez-Hilfiker, F., Liu, J., Shilling, J. E., Xing, J., Mathur, R., Middlebrook, A. M., Liao, J., Welti, A., Graus, M., Warneke, C., de Gouw, J. A., Holloway, J. S., Ryerson, T. B., Pollack, I. B., and Thornton, J. A.: Anthropogenic enhancements to production of highly oxygenated molecules from autoxidation, P. Natl. Acad. Sci. USA, 116, 6641-6646, 10.1073/pnas.1810774116, 2019.

Roldin, P., Ehn, M., Kurtén, T., Olenius, T., Rissanen, M. P., Sarnela, N., Elm, J., Rantala, P., Hao, L., Hyttinen, N., Heikkinen, L., Worsnop, D. R., Pichelstorfer, L., Xavier, C., Clusius, P., Öström, E., Petäjä, T., Kulmala, M., Vehkamäki, H., Virtanen, A., Riipinen, I., and Boy, M.: The role of highly oxygenated organic molecules in the Boreal aerosol-cloud-climate system, Nat.Commun., 10, 4370, 10.1038/s41467-019-12338-8, 2019.

Weber, J., Archer-Nicholls, S., Griffiths, P., Berndt, T., Jenkin, M., Gordon, H., Knote, C., and Archibald, A. T.: CRI-HOM: A novel chemical mechanism for simulating highly oxygenated organic

molecules (HOMs) in global chemistry–aerosol–climate models, Atmos. Chem. Phys., 20, 10889-10910, 10.5194/acp-20-10889-2020, 2020.

Xu, L., Pye, H. O. T., He, J., Chen, Y., Murphy, B. N., and Ng, N. L.: Experimental and model estimates of the contributions from biogenic monoterpenes and sesquiterpenes to secondary organic aerosol in the southeastern United States, Atmos. Chem. Phys., 18, 12613-12637, 10.5194/acp-18-12613-2018, 2018.

**Major Comment#2**: Additionally, Liu et al (2024) identifies the branching in the NO termination pathway as being highly uncertain, but here the sensitivity to the rate of NO reaction is investigated. Why was the sensitivity to the rate rather than the branching ratio studied?

**Response:** In Liu et al. (2024), the adjustment of the branching ratio from 0.4 to 0 in the NO termination pathway of HOMs is based on significant uncertainties in this parameter (Weber et al., 2020; Roldin et al., 2019; Xu et al., 2022). Although this adjustment indeed alleviates the overestimation of HOMs concentrations at the Centreville and SMEAR II site (Fig. 2 in Liu et al., 2024, see below), it results in non-conservation of carbon atoms between reactants and products since the reaction changes from "MT-HOM-$RO_2$ + NO → 0.8*$NO_2$ + 0.8*$HO_2$ + 0.4*SOAGhmb + 0.8*HYDRALD + 0.2*SOAGhmn" to "MT-HOM-$RO_2$ + NO → 0.8*$NO_2$ + 0.8*$HO_2$ + 0.8*HYDRALD + 0.2*SOAGhmn", where MT-HOM-$RO_2$ is the multi-generation products of autoxidation, SOAGhmb and SOAGhmn are the gas phase C10-HOMs and HYDRALD is the lumped unsaturated hydroxycarbonyl (details of this reaction are shown in Table S2 in the response of minor comments 3 and 13).

Consequently, considering the overestimation of NO concentrations by a factor of five at Centreville and SMEAR II station, we adjusted the rate of the NO termination pathway to one-fifth of its original value to assess its impact on HOM concentrations as well as NPF rate. Also, adjusting the reaction rate to 20% of the original value is almost equivalent to adjusting the branching ratio from 0.4 to 0.08, which is very close to the experiment designed by Liu et al. (2024) (the branching ratio of HOMs is set as zero).

[Figure]

Figure. 2 in Liu et al., 2024: The diurnal cycle of observed (dots) surface C10-ON (a, b) and C10-NON (c, d) concentrations (unit: ng/m3) at the Centreville site during the SENEX campaign (a, c) and the SMEAR II sites during the BAECC (b, d) campaign. The simulated surface C10 HOMs (C10-ON and C10-NON) concentrations at the closest grid to the Centreville and the SMEAR II sites are used from the addHOMs (solid lines) and no_HMB_NO (dashed lines) experiments. The simulated C10 HOMs at two sites are scaled by the ratios of the observed monoterpene concentrations to the simulated monoterpene concentrations. AddHOMs is the basic experiments when adding HOMs chemistry and partitioning and no_HMB_NO is the experiments adjusting the yield of SOAGhmb as zero in NO termination pathway.

**Reference**

Liu, Y., Dong, X., Wang, M., Xu, R., Thornton, J. A., Shao, X., Emmons, L. K., Jo, D. S., Yue, M., and Shrivastava, M.: A Modeling Study of Global Distribution and Formation Pathways of Highly Oxygenated Organic Molecules Derived Secondary Organic Aerosols (HOMs-SOA) from Monoterpenes, J. Geophys. Res.-Atmos., (under review), 2024.

Roldin, P., Ehn, M., Kurtén, T., Olenius, T., Rissanen, M. P., Sarnela, N., Elm, J., Rantala, P., Hao, L., Hyttinen, N., Heikkinen, L., Worsnop, D. R., Pichelstorfer, L., Xavier, C., Clusius, P., Öström, E., Petäjä, T., Kulmala, M., Vehkamäki, H., Virtanen, A., Riipinen, I., and Boy, M.: The role of highly oxygenated organic molecules in the Boreal aerosol-cloud-climate system, Nat.Commun., 10, 4370, 10.1038/s41467-019-12338-8, 2019.

Weber, J., Archer-Nicholls, S., Griffiths, P., Berndt, T., Jenkin, M., Gordon, H., Knote, C., and Archibald, A. T.: CRI-HOM: A novel chemical mechanism for simulating highly oxygenated organic molecules (HOMs) in global chemistry–aerosol–climate models, Atmos. Chem. Phys., 20, 10889-10910, 10.5194/acp-20-10889-2020, 2020.

Xu, R. C., Thornton, J. A., Lee, B., Zhang, Y. X., Jaegle, L., Lopez-Hilfiker, F. D., Rantala, P., and Petaja, T.: Global simulations of monoterpene-derived peroxy radical fates and the distributions of highly oxygenated organic molecules (HOMs) and accretion products, Atmos. Chem. Phys., 22, 5477-5494, 10.5194/acp-22-5477-2022, 2022.

**Major Comment#3 and #4**: In Table 1, it is shown that all C20 and C15 compounds are represented by one species each with one volatility each. As seen in previous work (Stolzenburg 2018, Ye et al 2018, Schervish and Donahue 2020, etc.) not all accretion products lead to ULVOCs or even ELVOCs. The assumption that they do seems like it would dramatically overestimate the role of organic nucleation and growth.

While the low branching ratio to accretion products may somewhat account for the concern brought up in point 3, experimentally many C20s end up in the E/LVOC range, allowing them to contribute to small particle growth, and the mechanism seems to indicate products from C10+C10 accretion reactions can only be C20 ULVOCs or non-HOM species.

**Response:** Yes, the lack of consideration for C15 and C20 dimers in LVOCs and C20 dimers in ELVOCs is a limitation of the current chemical mechanism we used in CAM6-Chem. We appreciate the referee for providing some articles which showed the molecular formulas and concentration of accretion product in different volatility bins. But the explicit chemical kinetics of related reactions (i.e. the intermediate products and their yields) are not displayed. Therefore, we are unable to represent all the final products mentioned in these articles in the CAM6-Chem model.

However, we realize that this uncertainty is important and should be thoroughly discussed. Therefore, we have added the following discussion at the end of the **Summary**:

"There might be some overestimations with C15 and C20 involved in new particle formation if we assume that all the accretion products are ELVOC or ULVOC. In the updated model, $C_{15}H_{18}O_9$ (C15, extremely low volatility) and $C_{20}H_{32}O_8$ (C20, ultra-low volatility) are just simplified representatives of all C15 and C20 dimers. Although more dimer species with low volatility has been already detected on chamber experiments (Stolzenburg et al., 2018; Ye et al., 2018; Schervish and Donahue, 2020), they did not provide the explicit chemical kinetics of related reactions (i.e. the intermediate products and their yields). On the other hand, although yields of accretion products vary by 1 to 2 orders of magnitude in previous studies (Rissanen et al., 2015; Berndt et al., 2018; Zhao et al., 2018), the yields of C15 and C20 we currently use are very low (4%), resulting in relatively low dimer concentrations.

Even if they were all ELVOC and ULVOC, it would not lead to significant overestimation, and therefore, would not substantially impact nucleation and growth rates."

**Reference**

Berndt, T., Mentler, B., Scholz, W., Fischer, L., Herrmann, H., Kulmala, M., and Hansel, A.: Accretion Product Formation from Ozonolysis and OH Radical Reaction of α-Pinene: Mechanistic Insight and the Influence of Isoprene and Ethylene, Environ. Sci. Technol., 52, 11069-11077, 10.1021/acs.est.8b02210, 2018.

Rissanen, M. P., Kurtén, T., Sipilä, M., Thornton, J. A., Kausiala, O., Garmash, O., Kjaergaard, H. G., Petäjä, T., Worsnop, D. R., Ehn, M., and Kulmala, M.: Effects of Chemical Complexity on the Autoxidation Mechanisms of Endocyclic Alkene Ozonolysis Products: From Methylcyclohexenes toward Understanding α-Pinene, J. Phys. Chem. A, 119, 4633-4650, 10.1021/jp510966g, 2015.

Schervish, M. and Donahue, N. M.: Peroxy radical chemistry and the volatility basis set, Atmospheric Chemistry and Physics, 20, 1183-1199, 10.5194/acp-20-1183-2020, 2020.

Stolzenburg, D., Fischer, L., Vogel, A. L., Heinritzi, M., Schervish, M., Simon, M., Wagner, A. C., Dada, L., Ahonen, L. R., Amorim, A., Baccarini, A., Bauer, P. S., Baumgartner, B., Bergen, A., Bianchi, F., Breitenlechner, M., Brilke, S., Buenrostro Mazon, S., Chen, D., Dias, A., Draper, D. C., Duplissy, J., El Haddad, I., Finkenzeller, H., Frege, C., Fuchs, C., Garmash, O., Gordon, H., He, X., Helm, J., Hofbauer, V., Hoyle, C. R., Kim, C., Kirkby, J., Kontkanen, J., Kürten, A., Lampilahti, J., Lawler, M., Lehtipalo, K., Leiminger, M., Mai, H., Mathot, S., Mentler, B., Molteni, U., Nie, W., Nieminen, T., Nowak, J. B., Ojdanic, A., Onnela, A., Passananti, M., Petäjä, T., Quéléver, L. L. J., Rissanen, M. P., Sarnela, N., Schallhart, S., Tauber, C., Tomé, A., Wagner, R., Wang, M., Weitz, L., Wimmer, D., Xiao, M., Yan, C., Ye, P., Zha, Q., Baltensperger, U., Curtius, J., Dommen, J., Flagan, R. C., Kulmala, M., Smith, J. N., Worsnop, D. R., Hansel, A., Donahue, N. M., and Winkler, P. M.: Rapid growth of organic aerosol nanoparticles over a wide tropospheric temperature range, P. Natl. Acad. Sci. USA, 115, 9122-9127, 10.1073/pnas.1807604115, 2018.

Ye, Q., Wang, M., Hofbauer, V., Stolzenburg, D., Chen, D., Schervish, M., Vogel, A., Mauldin, R. L., Baalbaki, R., Brilke, S., Dada, L., Dias, A., Duplissy, J., El Haddad, I., Finkenzeller, H., Fischer, L., He, X., Kim, C., Kürten, A., Lamkaddam, H., Lee, C. P., Lehtipalo, K., Leiminger, M., Manninen, H. E., Marten, R., Mentler, B., Partoll, E., Petäjä, T., Rissanen, M., Schobesberger, S., Schuchmann, S., Simon, M., Tham, Y. J., Vazquez-Pufleau, M., Wagner, A. C., Wang, Y., Wu, Y., Xiao, M., Baltensperger, U., Curtius, J., Flagan, R., Kirkby, J., Kulmala, M., Volkamer, R., Winkler, P. M., Worsnop, D., and Donahue, N. M.: Molecular Composition and Volatility of Nucleated Particles from

α-Pinene Oxidation between −50 °C and +25 °C, Environ. Sci. Technol., 53, 12357-12365, 10.1021/acs.est.9b03265, 2019.

Zhao, Y., Thornton, J. A., and Pye, H. O. T.: Quantitative constraints on autoxidation and dimer formation from direct probing of monoterpene-derived peroxy radical chemistry, P. Natl. Acad. Sci. USA, 115, 12142-12147, 10.1073/pnas.1812147115, 2018.

**Major Comment#5**: Why are only 2 steps of autoxidation simulated? Laboratory evidence (Heinritzi et al 2020, Simon et al 2020, etc.) shows products with very high oxygen content that likely underwent more than 2 steps of autoxidation. Would allowing more autoxidation lead to a higher organic contribution to nucleation, or perhaps this model step up could suggest how many steps are likely to occur in the actual atmosphere prior to termination or condensation.

**Response:** Currently, two-step autoxidation reactions are used to approximately represent the formation of C10-HOMs undergoing multiple steps of autoxidation reactions (more than one step). As mentioned in Heinritzi et al. (2020), three steps of autoxidation products were reported (i.e., $C_{10}H_{15}O_{4, 6, 8, 10}$ radicals) in chamber experiments yet more than three generations might occur. Additionally, Heinritzi et al. (2020) and Simon et al. (2020) did not provide the chemical reaction and reaction rates for multi-step oxidation products. So in the model we use two steps to approximate multi-generation.

We appreciate the referee for reminding on this important issue and this uncertainty should be thoroughly discussed, so we have added the following discussion at the end of the **Summary**:

[revised manuscript text omitted]

Kurtén, T., Tiusanen, K., Roldin, P., Rissanen, M., Luy, J.-N., Boy, M., Ehn, M., and Donahue, N.: α-Pinene Autoxidation Products May Not Have Extremely Low Saturation Vapor Pressures Despite High O:C Ratios, J. Phys. Chem. A, 120, 2569-2582, 10.1021/acs.jpca.6b02196, 2016.

Simon, M., Dada, L., Heinritzi, M., Scholz, W., Stolzenburg, D., Fischer, L., Wagner, A. C., Kürten, A., Rörup, B., He, X. C., Almeida, J., Baalbaki, R., Baccarini, A., Bauer, P. S., Beck, L., Bergen, A., Bianchi, F., Bräkling, S., Brilke, S., Caudillo, L., Chen, D., Chu, B., Dias, A., Draper, D. C., Duplissy, J., El-Haddad, I., Finkenzeller, H., Frege, C., Gonzalez-Carracedo, L., Gordon, H., Granzin, M., Hakala, J., Hofbauer, V., Hoyle, C. R., Kim, C., Kong, W., Lamkaddam, H., Lee, C. P., Lehtipalo, K., Leiminger, M., Mai, H., Manninen, H. E., Marie, G., Marten, R., Mentler, B., Molteni, U., Nichman, L., Nie, W., Ojdanic, A., Onnela, A., Partoll, E., Petäjä, T., Pfeifer, J., Philippov, M., Quéléver, L. L. J., Ranjithkumar, A., Rissanen, M. P., Schallhart, S., Schobesberger, S., Schuchmann, S., Shen, J., Sipilä, M., Steiner, G., Stozhkov, Y., Tauber, C., Tham, Y. J., Tomé, A. R., Vazquez-Pufleau, M., Vogel, A. L., Wagner, R., Wang, M., Wang, D. S., Wang, Y., Weber, S. K., Wu, Y., Xiao, M., Yan, C., Ye, P., Ye, Q., Zauner-Wieczorek, M., Zhou, X., Baltensperger, U., Dommen, J., Flagan, R. C., Hansel, A., Kulmala, M., Volkamer, R., Winkler, P. M., Worsnop, D. R., Donahue, N. M., Kirkby, J., and Curtius, J.: Molecular understanding of new-particle formation from α-pinene between −50 and +25 °C, Atmos. Chem. Phys., 20, 9183-9207, 10.5194/acp-20-9183-2020, 2020.

Trostl, J., Chuang, W. K., Gordon, H., Heinritzi, M., Yan, C., Molteni, U., Ahlm, L., Frege, C., Bianchi, F., Wagner, R., Simon, M., Lehtipalo, K., Williamson, C., Craven, J. S., Duplissy, J., Adamov, A., Almeida, J., Bernhammer, A. K., Breitenlechner, M., Brilke, S., Dias, A., Ehrhart, S., Flagan, R. C., Franchin, A., Fuchs, C., Guida, R., Gysel, M., Hansel, A., Hoyle, C. R., Jokinen, T., Junninen, H., Kangasluoma, J., Keskinen, H., Kim, J., Krapf, M., Kurten, A., Laaksonen, A., Lawler, M., Leiminger, M., Mathot, S., Mohler, O., Nieminen, T., Onnela, A., Petaja, T., Piel, F. M., Miettinen, P., Rissanen, M. P., Rondo, L., Sarnela, N., Schobesberger, S., Sengupta, K., Sipila, M., Smith, J. N., Steiner, G., Tome, A., Virtanen, A., Wagner, A. C., Weingartner, E., Wimmer, D., Winkler, P. M., Ye, P. L., Carslaw, K. S., Curtius, J., Dommen, J., Kirkby, J., Kulmala, M., Riipinen, I., Worsnop, D. R., Donahue, N. M., and Baltensperger, U.: The role of low-volatility organic compounds in initial particle growth in the atmosphere, Nature, 533, 527-+, 10.1038/nature18271, 2016.

**Major Comment#6**: Are any particle-phase processing of HOMs considered such as particle phase oligomerization or decomposition of accretion products and organic hydroperoxides?

**Response:** Particle phase oligomerization or decomposition were not taken into considered in our updated model. One reason is that the default version of CAM-Chem did not include these process (Jo et al., 2023). Also, compared to particle phase, we are more focused on the process that may affect the concentration of gaseous HOMs. Additionally, the deposition and photolysis of particle phase HOMs were added in our model.

However, we think it is necessary to discuss the uncertainties arising from disregarding these processes, and we have added the following content to the discussion of uncertainties in the **Summary**:

"Neglecting the oligomerization of accretion products can lead to higher volatility of aerosols, resulting in reduced the mass concentration in the particle phase and reduced condensation sink (CS), but increased mass in the gaseous phase. This could lead to an overestimation of the NPF rate. However, since the mass of HOMs-SOA accounts for only about 10% of the total SOA mass, the impact on NPF rate can be neglected.

Not considering decomposition of accretion products may lead to an overestimation of the mass and number concentration of HOMs in particle phase, and consequently an overestimation of CS and underestimation of the NPF rate. However, C15-SOA and C20-SOA account for less than 4% of the total SOA (Liu et al., 2024), so the impact of ignoring the decomposition of accretion products is negligible."

In our model, organic hydroperoxides are in gaseous phase so we did not consider decomposition of organic hydroperoxides. Since the intermediate $RO_2$ lifetime will rarely exceed about 100 s (Bianchi et al., 2018), the impact of ignoring its decomposition on NPF rate is minimal.


**Response:** We agree with the referee that an explicit list of the chemical reactions of HOMs would enhance the detail and clarity of our manuscript. Currently, the paper by Liu et al. (2024) that contains a comprehensive description of the mechanism is still under review. Consequently, we prefer not to transpose the same content to this manuscript. Nevertheless, to address the concern raised by the reviewers regarding our submission, we have added a succinct description of the machniasm that captures the essential details to the supplementary material in this revision, with a schematic figure and a list of representative chemical reactions. Furthermore, we have prepared a more thorough account of the complete chemical mechanism as outlined below:

Figure S1 shows a flowchart of the HOMs mechanism implemented into CAM6-Chem. In general, monoterpenes are oxidized by OH radicals or $O_3$ to form MT-$aRO_2$ and MT-$bRO_2$ radicals. MT-$bRO_2$ undergo multi-step autoxidation reactions to form HOMs with 10 carbon atoms (C10-HOMs) (green arrows in Fig. S1). The intermediates for the two-step autoxidation are MT-$cRO_2$ and MT-HOM-$RO_2$. The MT-HOM-$RO_2$ radical represents the $RO_2$ radicals that undergo two or multi-step autoxidation. On the one hand, MT-HOM-$RO_2$ radicals are further oxidized to form C10-HOMs. On the other hand,

all the MT-RO$_2$ radicals (including MT-aRO$_2$, MT-bRO$_2$, MT-cRO$_2$, and MT-HOM-RO$_2$) undergo self- and cross-reactions to form accretion products (C15 and C20) (orange arrows in Fig. S1). The formation processes of C10-HOMs can be terminated by several oxidants (gray arrows in Fig. S1). SOA is formed via gas-particle partitioning processes of C10-HOMs, C15 and C20 (blue dashed arrows in Fig. S1).

[Figure]

Figure S1. The flow chart of the formation and gas-particle partitioning processes of HOMs and accretion products. The green arrows represent the autoxidation reactions. The gray curved solid arrows represent the termination reactions. The yellow arrows represent the self- and cross-reactions. The blue arrows represent the conversion between C10-CBYL\C10-ROH and MT-RO$_2$ radicals. The blue dashed arrows represent the gas-particle partitioning processes.

The formation, photolysis, and scavenging processes of C10-HOMs C15 and C20 are detailed discussed as follows.

1. Monoterpene oxidation and autoxidation

Reaction 1-8 show the new branching which forms MT-bRO$_2$ that can undergo autoxidation are included in the original monoterpene + OH\O$_3$ reactions. The APINO$_2$, BPINO$_2$, LIMONO$_2$, and MYRCO$_2$ (MT-aRO$_2$) are the original formed RO$_2$ that cannot form HOMs. Reaction 9-10 show the MT-bRO$_2$ may go through two generations of autoxidation reaction (Table S1).

Table S1. Initial oxidation between monoterpenes and OH radical

| Index | Reactions |
|-------|-----------|
| 1 | $APIN + OH \rightarrow 0.25*APINO_2 + 0.75*MT\text{-}bRO_2$ |
| 2 | $BPIN + OH \rightarrow 0.25*BPINO_2 + 0.75*MT\text{-}bRO_2$ |
| 3 | $LIMON + OH \rightarrow 0.25*LIMONO_2 + 0.75*MT\text{-}bRO_2$ |
| 4 | $MYRC + OH \rightarrow 0.25*MYRCO_2 + 0.75*MT\text{-}bRO_2$ |
| 5 | $APIN + O_3 \rightarrow$
$0.736*APINO_2 + 0.064*MT\text{-}bRO_2 + 0.77*OH + 0.066*TERPA2O_2 + 0.22*H_2O_2 + 0.044*TERPA +$
$0.002*TERPACID + 0.034*TERPA2 + 0.17*HO_2 + 0.17*CO + 0.27*CH_2O + 0.054*TERPA2CO_3$ |
| 6 | $BPIN + O_3 \rightarrow$
$0.736*BPINO_2 + 0.064*MT\text{-}bRO_2 + 0.102*TERPK + 0.3*OH + 0.06*TERPA2CO_3 + 0.32*H_2O_2 +$
$0.038*BIGALK + 0.19*CO_2 + 0.81*CH_2O + 0.11*HMHP + 0.08*HCOOH$ |
| 7 | $LIMON + O_3 \rightarrow$
$0.736*LIMONO_2 + 0.064*MT\text{-}bRO_2 + 0.66*OH + 0.132*TERPF1 + 0.33*CH_3CO_3 + 0.33*CH_2O +$
$0.066*TERPA3CO_3 + 0.33*H_2O_2 + 0.002*TERPACID$ |
| 8 | $MYRC + O_3 \rightarrow$
$0.736*MYRCO_2 + 0.064*MT\text{-}bRO_2 + 0.2*TERPF2 + 0.63*OH + 0.63*HO_2 + 0.25*CH_3COCH_3$
$+0.39*CH_2O + 0.18*HYAC$ |
| 9 | $MT\text{-}bRO_2 \rightarrow MT\text{-}cRO_2$ |
| 10 | $MT\text{-}cRO_2 \rightarrow MT\text{-}HOM\text{-}RO_2$ |

**2. Formation of C10-HOMs and accretion products**

Reaction 11-24 show self- and cross-reactions of MT-RO$_2$ and ISOP-RO$_2$ to form accretion products (SOAGac15 and SOAGac20). Reaction 25-27 show the MT-HOM-RO$_2$ are oxidized by HO$_2$\NO\NO$_3$ to form C10-HOMs, including non-nitrate HOMs (SOAGhma and SOAGhmb) and nitrate HOMs (SOAGhmn) (Table S2).

Table S2. Self- and cross-reactions to form gas-phase accretion products

| Index | Reactions |
|-------|-----------|
| 11 | $MT\text{-}aRO_2 + MT\text{-}aRO_2 \rightarrow$
$0.893*C_{10}\text{-}CBYL + 0.29*C_{10}\text{-}ROH + 0.603*HO_2 + 1.34*HYDRALD + 0.067*MT\text{-}bRO_2 + 0.04*SOAGac20$ |
| 12 | $MT\text{-}aRO_2 + MT\text{-}bRO_2 \rightarrow$
$0.96*C_{10}\text{-}CBYL + 0.29*C_{10}\text{-}ROH + 0.67*HO_2 + 1.34*HYDRALD + 0.04*SOAGac20$ |

| 13 | MT-aRO$_2$ + MT-cRO$_2$ → |
| | 0.96*C$_{10}$-CBYL + 0.29*C$_{10}$-ROH + 0.67*HO$_2$ + 1.34*HYDRALD + 0.04*SOAGac20 |
| 14 | MT-aRO$_2$ + MT-HOM-RO$_2$ → |
| | 0.96*C$_{10}$-CBYL + 0.29*C$_{10}$-ROH + 0.67*HO$_2$ + 1.34*HYDRALD + 0.04*SOAGac20 |
| 15 | MT-aRO$_2$ + ISOP-RO$_2$ → |
| | 0.4465*C$_{10}$-CBYL + 0.145*C$_{10}$-ROH + 0.145*ROH + 0.603*HO$_2$ + 1.485*HYDRALD + 0.0335*MT-bRO$_2$ |
| | + 0.04*SOAGac15 |
| 16 | MT-bRO$_2$ + MT-bRO$_2$ → |
| | 0.96*C$_{10}$-CBYL + 0.29*C$_{10}$-ROH + 0.67*HO$_2$ + 1.34*HYDRALD + 0.04*SOAGac20 |
| 17 | MT-cRO$_2$ + MT-cRO$_2$ → |
| | 0.96*C$_{10}$-CBYL + 0.29*C$_{10}$-ROH + 0.67*HO$_2$ + 1.34*HYDRALD + 0.04*SOAGac20 |
| 18 | MT-HOM-RO$_2$ + MT-HOM-RO$_2$ → |
| | 0.96*C$_{10}$-CBYL + 0.29*C$_{10}$-ROH + 0.67*HO$_2$ + 1.34*HYDRALD + 0.04*SOAGac20 |
| 19 | MT-bRO$_2$ + MT-cRO$_2$ → |
| | 0.96*C$_{10}$-CBYL + 0.29*C$_{10}$-ROH + 0.67*HO$_2$ + 1.34*HYDRALD + 0.04*SOAGac20 |
| 20 | MT-bRO$_2$ + MT-HOM-RO$_2$ → |
| | 0.96*C$_{10}$-CBYL + 0.29*C$_{10}$-ROH + 0.67*HO$_2$ + 1.34*HYDRALD + 0.04*SOAGac20 |
| 21 | MT-cRO$_2$ + MT-HOM-RO$_2$ → |
| | 0.96*C$_{10}$-CBYL + 0.29*C$_{10}$-ROH + 0.67*HO$_2$ + 1.34*HYDRALD + 0.04*SOAGac20 |
| 22 | MT-bRO$_2$ + ISOP-RO$_2$ → |
| | 0.48*C$_{10}$-CBYL + 0.145*C$_{10}$-ROH + 0.145*ROH + 0.67*HO$_2$ + 1.485*HYDRALD + 0.04*SOAGac15 |
| 23 | MT-cRO$_2$ + ISOP-RO$_2$ → |
| | 0.48*C$_{10}$-CBYL + 0.145*C$_{10}$-ROH + 0.145*ROH + 0.67*HO$_2$ + 1.485*HYDRALD + 0.04*SOAGac15 |
| 24 | MT-HOM-RO$_2$ + ISOP-RO$_2$ → |
| | 0.48*C$_{10}$-CBYL + 0.145*C$_{10}$-ROH + 0.145*ROH + 0.67*HO$_2$ + 1.485*HYDRALD + 0.04*SOAGac15 |
| 25 | MT-HOM-RO$_2$ + HO$_2$ → SOAGhma + O$_2$ |
| 26 | MT-HOM-RO$_2$ + NO → |
| | 0.8*NO$_2$ + 0.8*HO$_2$ + 0.4*SOAGhmb + 0.8*HYDRALD + 0.2*SOAGhmn |
| 27 | MT-HOM-RO$_2$ + NO$_3$ → HO$_2$ + NO$_2$ + 0.5*SOAGhmb + HYDRALD |

3. Other reactions of MT-RO$_2$

Reaction 28-37 show MT-bRO$_2$, MT-cRO$_2$, and MT-HOM-RO$_2$ are terminated by methylperoxy/peroxyacetyl radicals. Reaction 38-49 show the MT-bRO$_2$\MT-cRO$_2$ reacts with NO/NO$_3$ (Table S3).

Table S3. MT-RO$_2$ reactions with methylperoxy/peroxyacetyl radicals

| Index | Reactions |
| --- | --- |
| 28 | APINO$_2$ + CH$_3$CO$_3$ → |

| | |
|---|---|
| | $0.05*MT\text{-}bRO_2 + 0.3705*TERPA + 0.3325*TERPA3 + 0.133*TERP1OOH + 0.12*CH_3COCH_3 + 0.114*TERPF1 + 0.27*CH_2O + HO_2 + CH_3O_2 + CO_2$ |
| 29 | $APINO_2 + CH_3O_2 \rightarrow$
 $0.05*MT\text{-}bRO_2 + 0.83*CH_2O + 0.133*TERPF1 + 0.399*TERPA + 0.19*TERPA3 + 0.1235*TERP1OOH + 0.17*CH_3OH + 0.1045*TERPK + 0.06*CH_3COCH_3 + 1.16*HO_2$ |
| 30 | $BPINO_2 + CH_3CO_3 \rightarrow$
 $0.05*MT\text{-}bRO_2 + 0.304*TERPK + 0.2565*TERPF1 + 0.3895*TERPA3 + 0.11*CH_3COCH_3 + 0.65*CH_2O + HO_2 + CH_3O_2 + CO_2$ |
| 31 | $BPINO_2 + CH_3O_2 \rightarrow$
 $0.05*MT\text{-}bRO_2 + 1.4*CH_2O + 0.3515*TERPF1 + 0.304*TERPK + 1.5*HO_2 + 0.08*CH_3COCH_3 + 0.2945*TERPA3$ |
| 32 | $LIMONO_2 + CH_3CO_3 \rightarrow$
 $0.05*MT\text{-}bRO_2 + 0.95*TERPF1 + 0.56*CH_2O + HO_2 + CH_3O_2 + CO_2$ |
| 33 | $LIMONO_2 + CH_3O_2 \rightarrow$
 $0.05*MT\text{-}bRO_2 + 0.25*CH_3OH + 0.95*TERPF1 + 1.03*CH_2O + HO_2$ |
| 34 | $MYRCO_2 + CH_3CO_3 \rightarrow$
 $0.05*MT\text{-}bRO_2 + 0.95*TERPF2 + HO_2 + 0.46*CH_3COCH_3 + 0.42*CH_2O + CH_3O_2 + CO_2$ |
| 35 | $MYRCO_2 + CH_3O_2 \rightarrow$
 $0.05*MT\text{-}bRO_2 + 0.25*CH_3OH + 0.95*TERPF2 + 0.75*CH_2O + HO_2$ |
| 36 | $MT\text{-}bRO_2 \setminus MT\text{-}cRO_2 \setminus MT\text{-}HOM\text{-}RO_2 + CH_3O_2 \rightarrow$
 $0.15*CH_3OH + 0.85*CH_2O + 1.4*HO_2 + 0.7*HYDRALD + 0.7*CH_3COCH_3 + 0.15*C_{10}\text{-}ROH + 0.15*C_{10}\text{-}CBYL$ |
| 37 | $MT\text{-}bRO_2 \setminus MT\text{-}cRO_2 \setminus MT\text{-}HOM\text{-}RO_2 + CH_3CO_3 \rightarrow$
 $0.7*CH_3O_2 + 0.7*HO_2 + 0.7*HYDRALD + 0.7*CH_3COCH_3 + 0.3*CH_3COOH + 0.15*C_{10}\text{-}ROH + 0.15*C_{10}\text{-}CBYL$ |
| 38 | $APINO_2 + NO \rightarrow$
 $0.05*MT\text{-}bRO_2 + 0.0095*TERPHFN + 0.019*TERPNS1 + 0.095*TERPNS + 0.0475*TERPNT + 0.0475*TERPNT1 + 0.77*NO_2 + 0.77*HO_2 + 0.285*TERPA + 0.2565*TERPA3 + 0.09*CH_3COCH_3 + 0.0855*TERPF1 + 0.21*CH_2O + 0.1045*TERP1OOH$ |
| 39 | $APINO_2 + NO_3 \rightarrow$
 $0.05*MT\text{-}bRO_2 + NO_2 + HO_2 + 0.3705*TERPA + 0.3325*TERPA3 + 0.12*CH_3COCH_3 + 0.114*TERPF1 + 0.27*CH_2O + 0.133*TERP1OOH$ |
| 40 | $BPINO_2 + NO \rightarrow$
 $0.05*MT\text{-}bRO_2 + 0.08*CH_3COCH_3 + 0.49*CH_2O + 0.19*TERPF1 + 0.228*TERPK + 0.038*TERPNS1 + 0.019*TERPNS + 0.057*TERPNT + 0.1235*TERPNT1 + 0.2945*TERPA3 + 0.75*HO_2 + 0.75*NO_2$ |
| 41 | $BPINO_2 + NO_3 \rightarrow$
 $0.05*MT\text{-}bRO_2 + 0.11*CH_3COCH_3 + 0.65*CH_2O + 0.2565*TERPF1 + 0.304*TERPK + 0.3895*TERPA3 + HO_2 + NO_2$ |
| 42 | $LIMONO_2 + NO \rightarrow$
 $0.05*MT\text{-}bRO_2 + 0.1615*TERPNT1 + 0.057*TERPNS1 + 0.77*NO_2 + 0.7315*TERPF1 + 0.77*HO_2 + 0.43*CH_2O$ |
| 43 | $LIMONO_2 + NO_3 \rightarrow$
 $0.05*MT\text{-}bRO_2 + NO_2 + 0.95*TERPF1 + HO_2 + 0.56*CH_2$ |
| 44 | $MYRCO_2 + NO \rightarrow$ |

[Figure]

| Index | Reactions |
|---|---|
| | $0.05*MT\text{-}bRO_2 + 0.095*TERPNS1 + 0.1805*TERPNT1 + 0.71*NO_2 + 0.6745*TERPF2 +$ $0.33*CH_3COCH_3 + 0.3*CH_2O + 0.71*HO_2$ |
| 45 | $MYRCO_2 + NO_3 \rightarrow$ $0.05*MT\text{-}bRO_2 + NO_2 + 0.95*TERPF2 + 0.46*CH_3COCH_3 + 0.42*CH_2O + HO_2$ |
| 46 | $MT\text{-}bRO_2\backslash MT\text{-}cRO_2 + NO \rightarrow$ $0.01*TERPHFN + 0.02*TERPNS1 +$ $0.1*TERPNS + 0.05*TERPNT + 0.05*TERPNT1 + 0.77*NO_2 + 0.77*HO_2 + 0.3*TERPA + 0.27*TERPA3 +$ $0.09*CH_3COCH_3 + 0.09*TERPF1 + 0.21*CH_2O + 0.11*TERP1OOH$ |
| 47 | $MT\text{-}bRO_2 + NO_3 \rightarrow$ $HO_2 + NO_2 + 0.3*C_{10}\text{-}CBYL + 0.7*HYDRALD + 0.7*ROH$ |
| 48 | $MT\text{-}cRO_2 + NO_3 \rightarrow$ $HO_2 + NO_2 + 0.75*C_{10}\text{-}CBYL + 0.25*MT\text{-}HOM\text{-}RO_2$ |
| 49 | $MT\text{-}bRO_2\backslash MT\text{-}cRO_2 + HO_2 \rightarrow$ $0.06*CH_3COCH_3 + 0.06*TERPF1 + 0.08*CH_2O + 0.25*TERP1OOH + 0.48*HO_2 + 0.4*TERPOOH +$ $0.29*TERPA + 0.35*OH$ |

4. Sink of newly added species

Reaction 50-52 show the chemical loss of three kinds of intermediate products (C10-CBYL, C10-ROH, and ROH) by reacting with OH radical (Table S4). All the newly added SOAG and SOA follows the same deposition processes with the original SOA and SOAG in VBS approach. Only particle-phase $C_{10}$ HOMs undergo photolysis process and the photolysis rate is set as about 1/60 of $NO_2$ photolysis rate (Xu et al., 2022).

Table S4. Chemical loss of $C_{10}$-CBYL and $C_{10}$-ROH

| Index | Reactions |
|---|---|
| 50 | $C_{10}\text{-}CBYL + OH \rightarrow$ $0.125*APINO_2 + 0.125*BPINO_2 + 0.125*MYRCO_2 + 0.125*LIMONO_2 + 0.475*MT\text{-}bRO_2 + 0.025*MT\text{-}cRO_2$ |
| 51 | $C_{10}\text{-}ROH + OH \rightarrow$ $0.125*APINO_2 + 0.125*BPINO_2 + 0.125*MYRCO_2 + 0.125*LIMONO_2 + 0.475*MT\text{-}bRO_2 + 0.025*MT\text{-}cRO_2$ |
| 52 | $ROH + OH \rightarrow HO_2 + CH_3COCH_3$ |

Table S5. Species for HOMs and ACC formation mechanism.

| Species | Molecular formula | Description |
|---|---|---|
| APIN [b] | $C_{10}H_{16}$ | α-pinene |
| BPIN [b] | $C_{10}H_{16}$ | β-pinene |
| LIMON [b] | $C_{10}H_{16}$ | Limonene |
| MYRC [b] | $C_{10}H_{16}$ | Myrcene |
| APINO_2 [b] | $C_{10}H_{17}O_3$ | peroxy radical from OH +α-pinene reaction |

| | | |
|---|---|---|
| BPINO$_2$ [b] | C$_{10}$H$_{17}$O$_3$ | peroxy radical from OH + β-pinene reaction |
| LIMONO$_2$ [b] | C$_{10}$H$_{17}$O$_3$ | peroxy radical from OH + limonene |
| MYRCO$_2$ [b] | C$_{10}$H$_{17}$O$_3$ | peroxy radical from OH + myrcene |
| ISOPB1O$_2$ [b] | C$_5$H$_9$O$_3$ | OH-1-O$_2$-2—isoprene hydroxy peroxy radical |
| ISOPZD1O$_2$ [b] | C$_5$H$_9$O$_3$ | OH-1-O$_2$-4-Z—isoprene hydroxy peroxy radical |
| ISOPZD4O$_2$ [b] | C$_5$H$_9$O$_3$ | OH-4-O$_2$-1-Z—isoprene hydroxy peroxy radical |
| ISOPED1O$_2$ [b] | C$_5$H$_9$O$_3$ | OH-1-O$_2$-4-E—isoprene hydroxy peroxy radical |
| ISOPED4O$_2$ [b] | C$_5$H$_9$O$_3$ | OH-4-O$_2$-1-E—isoprene hydroxy peroxy radical |
| ISOPB4O$_2$ [b] | C$_5$H$_9$O$_3$ | OH-4-O$_2$-3—isoprene hydroxy peroxy radical |
| MT-bRO$_2$ [a] | C$_{10}$H$_{16}$O$_4$ | RO$_2$ from monoterpene+O$_3$/OH that can undergo autoxidation |
| MT-cRO$_2$ [a] | C$_{10}$H$_{16}$O$_6$ | RO$_2$ from MT-bRO$_2$ autoxidation |
| MT-HOM-RO$_2$ [a] | C$_{10}$H$_{16}$O$_8$ | RO$_2$ from MT-cRO$_2$ autoxidation |
| SOAGhma [a] | C$_{10}$H$_{14}$O$_9$ | gas-phase C10 HOMs product without nitrate from HO$_2$ reaction |
| SOAGhmb [a] | C$_{10}$H$_{14}$O$_9$ | gas-phase C10 HOMs product without nitrate from NO and NO$_3$ reaction |
| SOAGhmn [a] | C$_{10}$H$_{14}$O$_9$N | gas-phase C10 HOMs product with nitrate from NO reaction |
| SOAGac15 [a] | C$_{15}$H$_{18}$O$_7$ | gas-phase C15 accretion product from isoprene-derived RO$_2$ (ISOP-RO$_2$) + MT-RO$_2$ |
| SOAGac20 [a] | C$_{20}$H$_{32}$O$_8$ | gas-phase C20 accretion product from MT-RO$_2$ + MT-RO$_2$ |
| ROH [a] | C$_3$H$_8$O | lumped alcohols with more than 2 carbons |
| C$_{10}$-CBYL [a] | C$_{10}$H$_{17}$O$_3$ | Carbonyl with 10 carbon atoms |
| C$_{10}$-ROH [a] | C$_{10}$H$_{17}$O$_3$ | Alcohol with 10 carbon atoms |
| BIGALK | C$_5$H$_{12}$ | lumped alkanes C>3 |
| CH$_2$O [b] | CH$_2$O | formaldehyde |
| CH$_3$O$_2$ [b] | CH$_3$O$_2$ | methylperoxy radical |
| CH$_3$CO$_3$ [b] | CH$_3$CO$_3$ | acetylperoxy radical |
| CH$_3$COCH$_3$ [b] | CH$_3$COCH$_3$ | acetone |
| CH$_3$COOH [b] | CH$_3$COOH | acetic acid |
| CH$_3$OH [b] | CH$_3$OH | methanol |
| HO$_2$ [b] | HO$_2$ | hydroperoxyl radical |
| H$_2$O$_2$ [b] | H$_2$O$_2$ | hydrogen peroxide |
| HCOOH [b] | HCOOH | formic acid |
| HMHP [b] | CH$_4$O$_3$ | hydroxy methyl hydroperoxide |
| HYAC [b] | CH$_3$COCH$_2$OH | hydroxyacetone |
| HYDRALD [b] | HOCH$_2$CCH$_3$CHCHO | lumped unsaturated hydroxycarbonyl |
| TERP1OOH [b] | C$_{10}$H$_{18}$O$_3$ | terpene-derived hydroxy hydroperoxide with 1 double bond |
| TERPA [b] | C$_{10}$H$_{16}$O$_2$ | aldehyde terpene product with no double bonds that contains a ring like pinonaldehyde |
| TERPACID [b] | C$_{10}$H$_{16}$O$_4$ | carboxylic acid/peracid from TERPA |
| TERPA2 [b] | C$_9$H$_{14}$O$_2$ | TERPA oxidation product with no double bonds that contains an aldehydic group |
| TERPA2O$_2$ [b] | C$_9$H$_{15}$O$_4$ | TERPA peroxy radical 2nd step |
| TERPA2CO$_3$ [b] | C$_9$H$_{13}$O$_4$ | acyl peroxy radical from TERPA2 |
| TERPA3 [b] | C$_9$H$_{14}$O$_3$ | aldehyde terpene product with no ring like limonaldehyde |
| TERPA3CO$_3$ [b] | C$_9$H$_{13}$O$_5$ | acyl peroxy radical from TERPA3 |
| TERPF1 [b] | C$_{10}$H$_{16}$O$_2$ | functionalized terpene product with 1 double bond typically containing carbonyl groups |

| | | |
|---|---|---|
| TERPF2 [b] | $C_7H_{10}O$ | functionalized terpene product with 2 double bonds typically containing carbonyl groups |
| TERPHFN [b] | $C_{10}H_{19}NO_7$ | terpene highly functionalized nitrate |
| TERPK [b] | $C_9H_{14}O$ | terpene product containing a ketone group |
| TERPNS [b] | $C_{10}H_{17}NO_4$ | terpene-derived saturated secondary or primary nitrate |
| TERPNS1 [b] | $C_{10}H_{17}NO_4$ | terpene-derived unsaturated secondary or primary nitrate |
| TERPNT [b] | $C_{10}H_{17}NO_4$ | terpene-derived saturated tertiary nitrate |
| TERPNT1 [b] | $C_{10}H_{17}NO_4$ | terpene-derived unsaturated tertiary nitrate |

[a] Xu et al. (2022)

[b] Schwantes et al. (2020)

We chose the main chemical reactions and descriptions and then added them into the **Supplement**:

"Figure S1 shows a flowchart of the HOMs mechanism implemented into CAM6-Chem and Table S1 shows the main chemical reactions added into CAM6-Chem. In general, monoterpenes (including α-pinene, β-pinene, limonene and myrcene) are oxidized by OH radicals or $O_3$ to form MT-aRO$_2$ and MT-bRO$_2$ radicals (reactions 1-2 listed in Table S1, with only reactions involving α-pinene shown as an example). MT-bRO$_2$ undergo multi-step autoxidation reactions to form HOMs with 10 carbon atoms (C10-HOMs) (green arrows in Fig. S1 and reactions 3-4 in Table S1). The intermediates for the autoxidation are MT-cRO$_2$ and MT-HOM-RO$_2$. The MT-HOM-RO$_2$ radical represents the RO$_2$ radicals that undergo two or multi-step autoxidation. On the one hand, MT-HOM-RO$_2$ radicals are further oxidized to form C10-HOMs (reaction 8-10 in Table S1). On the other hand, all the MT-RO$_2$ radicals (including MT-aRO$_2$, MT-bRO$_2$, MT-cRO$_2$, and MT-HOM-RO$_2$) undergo self- and cross-reactions (orange arrows in Fig. S1) to form accretion products (C15 and C20) (reactions 5-7 in Table S1, with only reactions involving MT-aRO$_2$ shown as an example). The formation processes of C10-HOMs can be terminated by several oxidants (gray arrows in Fig. S1). SOA is formed via gas-particle partitioning processes of C10-HOMs, C15 and C20 (blue dashed arrows in Fig. S1S1). "

[Figure]

Figure S1. The flow chart of the formation and gas-particle partitioning processes of HOMs and accretion products. The green arrows represent the autoxidation reactions. The gray curved solid arrows represent the termination reactions. The yellow arrows represent the self- and cross-reactions. The blue arrows represent the conversion between C10-CBYL\C10-ROH and MT-RO$_2$ radicals. The blue dashed arrows represent the gas-particle partitioning processes.

Table S1. Main chemical reactions added in CAM6-Chem

| Index | Reactions |
|:---:|:---:|
| 1 | APIN + OH → 0.25*APINO$_2$ + 0.75*MT-bRO$_2$ |
| 2 | APIN + O$_3$ →
 0.736*APINO$_2$ + 0.064*MT-bRO$_2$ + 0.77*OH + 0.066*TERPA2O$_2$ + 0.22*H$_2$O$_2$ + 0.044*TERPA + 0.002*TERPACID +
 0.034*TERPA2 + 0.17*HO$_2$ + 0.17*CO + 0.27*CH$_2$O + 0.054*TERPA2CO$_3$ |
| 3 | MT-bRO$_2$ → MT-cRO$_2$ |
| 4 | MT-cRO$_2$ → MT-HOM-RO$_2$ |
| 5 | MT-aRO$_2$ + MT-aRO$_2$ →
 0.893*C$_{10}$-CBYL + 0.29*C$_{10}$-ROH + 0.603*HO$_2$ + 1.34*HYDRALD + 0.067*MT-bRO$_2$ + 0.04*SOAGac20 |
| 6 | MT-aRO$_2$ + MT-bRO$_2$ →
 0.96*C$_{10}$-CBYL + 0.29*C$_{10}$-ROH + 0.67*HO$_2$ + 1.34*HYDRALD + 0.04*SOAGac20 |
| 7 | MT-aRO$_2$ + ISOP-RO$_2$ →
 0.4465*C$_{10}$-CBYL + 0.145*C$_{10}$-ROH + 0.145*ROH + 0.603*HO$_2$ + 1.485*HYDRALD + 0.0335*MT-bRO$_2$ + 0.04*SOAGac15 |
| 8 | MT-HOM-RO$_2$ + HO$_2$ → SOAGhma + O$_2$ |
| 9 | MT-HOM-RO$_2$ + NO → |

| | |
|---|---|
| | $0.8*NO_2 + 0.8*HO_2 + 0.4*SOAGhmb + 0.8*HYDRALD + 0.2*SOAGhmn$ |
| 10 | MT-HOM-RO$_2$ + NO$_3$ → |
| | $HO_2 + NO_2 + 0.5*SOAGhmb + HYDRALD$ |

Table S2. Species for HOMs and ACC formation mechanism.

| Species | Molecular formula | Description |
|---|---|---|
| APIN [b] | $C_{10}H_{16}$ | α-pinene |
| APINO$_2$ [b] | $C_{10}H_{17}O_3$ | peroxy radical from OH +α-pinene reaction |
| MT-bRO$_2$ [a] | $C_{10}H_{16}O_4$ | RO$_2$ from monoterpene+O$_3$/OH that can undergo autoxidation |
| MT-cRO$_2$ [a] | $C_{10}H_{16}O_6$ | RO$_2$ from MT-bRO$_2$ autoxidation |
| MT-HOM-RO$_2$ [a] | $C_{10}H_{16}O_8$ | RO$_2$ from MT-cRO$_2$ autoxidation |
| SOAGhma [a] | $C_{10}H_{14}O_9$ | gas-phase C10 HOMs product without nitrate from HO$_2$ reaction |
| SOAGhmb [a] | $C_{10}H_{14}O_9$ | gas-phase C10 HOMs product without nitrate from NO and NO$_3$ reaction |
| SOAGhmn [a] | $C_{10}H_{14}O_9N$ | gas-phase C10 HOMs product with nitrate from NO reaction |
| SOAGac15 [a] | $C_{15}H_{18}O_7$ | gas-phase C15 accretion product from isoprene-derived RO$_2$ (ISOP-RO$_2$) + MT-RO$_2$ |
| SOAGac20 [a] | $C_{20}H_{32}O_8$ | gas-phase C20 accretion product from MT-RO$_2$ + MT-RO$_2$ |
| ROH [a] | $C_3H_8O$ | lumped alcohols with more than 2 carbons |
| C$_{10}$-CBYL [a] | $C_{10}H_{17}O_3$ | Carbonyl with 10 carbon atoms |
| C$_{10}$-ROH [a] | $C_{10}H_{17}O_3$ | Alcohol with 10 carbon atoms |
| CH$_2$O [b] | $CH_2O$ | formaldehyde |
| HO$_2$ [b] | $HO_2$ | hydroperoxyl radical |
| H$_2$O$_2$ [b] | $H_2O_2$ | hydrogen peroxide |
| HYDRALD [b] | $HOCH_2CCH_3CHCHO$ | lumped unsaturated hydroxycarbonyl |
| TERPA [b] | $C_{10}H_{16}O_2$ | aldehyde terpene product with no double bonds that contains a ring like pinonaldehyde |
| TERPACID [b] | $C_{10}H_{16}O_4$ | carboxylic acid/peracid from TERPA |
| TERPA2 [b] | $C_9H_{14}O_2$ | TERPA oxidation product with no double bonds that contains an aldehydic group |
| TERPA2O$_2$ [b] | $C_9H_{15}O_4$ | TERPA peroxy radical 2$^{nd}$ step |
| TERPA2CO$_3$ [b] | $C_9H_{13}O_4$ | acyl peroxy radical from TERPA2 |


**Minor Comment#6**: Paragraph at line 260: Can this analysis be made clearer? Why does an overestimation of $H_2SO_4$ lead to and underestimation in growth rate overall, but then also explains overestimated growth rates in specific cities?

**Response: Response:** We apologize for the unclear description. We have added the explanation at line 261:

"The underestimation of the sub-20nm growth rate in Inorg is due to an almost zero nucleation rate at around 1nm. Consequently, the absence of a nucleation rate results in the absence of NPF events and, thus, a zero growth rate. In contrast, in Inorg_Org, the NPF frequency is simulated accurately compared to that in Inorg (Fig, 1c). One contributing factor to the overestimation of the growth rate in Inorg_Org is the overestimation of the $H_2SO_4$ concentration, a feature of CAM6, as evidenced by comparisons with previous model simulations (Table S5) and measurements (Table S6)."

**Minor Comment#7**: Over- or underestimations in condensation sink are occasionally used to justify a corresponding under- or overestimation in NPF. This makes sense, however, can CS be prescribed in order to validate this?

**Response:** We are unable to implement this because the condensation sink (CS) is calculated online in the model. If we were to use a prescribed value, it would require data to be input every half hour (as the physical timestep of CAM6-Chem is half an hour). However, CS derived from measurements cannot provide such high temporal precision.

**Minor Comment#8**: Figure 4c, d can a label and the units be places under the x-axis?

**Response:** Thanks for reminding. We added the unit of $N_{20}$ and CCN ($cm^{-3}$) under the x-axis. The new figure is shown as follow.

[Figure]

**Minor Comment#9**: Line 340: Is the "organic nucleation rate" mentioned here just the sum of the neutral ($J_{Org,n}$) and ion-induced pure organic nucleation ($J_{Org,i}$) or does it also include inorganic-organic nucleation ($J_{SA-Org}$)?

**Response:** Yes. We apologize for the missing information in that sentence. The We have modified the sentence as follows.

"Globally, the vertically-integrated (below 15 km) annual mean organic nucleation rate $(J_{Org,n}+J_{Org,i}+J_{SA-Org})$ in Inorg_Org is $32 \times 10^6$ cm$^{-2}$ s$^{-1}$ (Fig. 5a), closely matching the inorganic nucleation rate of $39 \times 10^6$ cm$^{-2}$ s$^{-1}$ (Table 4)."

**Minor Comment#10 and # 11**: Figure 9: There are 2 panels labeled g.

Figure 9: There is only 1 unit given in the caption, but the left and right plots appear to have different units.

**Response:** We apologize for the spelling error and missing information in Figure 9. The new version of Figure 9 is as follow:

[Figure]

**Figure 9: Absolute differences (Units: cm$^{-2}$) and relative differences (Units: unitless) of in total vertically-integrated aerosol numbers in July 2013 between Inorg_Org and other sensitivity tests. Global mean values are shown on the top right of each figure. Model experiments are described in Table 2.**

We have added a new **Section 5** to analyze the impact of uncertainties from HOMs chemistry on aerosol and CCN numbers. Consequently, the discussions regarding the differences from Low_Br, as well as Slow_NO and Inorg_Org, have been moved to Section 5.

**Minor Comment#12**: Figure 10: There is reference in the caption to up to panel h, but the figure only contains up to panel d.

**Response:** We apologize for the wrong information in the caption and have corrected this. Now Figure 10 is as follow:

[Figure]

**Figure 10: Spatial distribution of annual mean total vertically-integrated CCN concentrations at 0.5% supersaturation for (a) Inorg and (c) Inorg_Org (unit: cm⁻²). Also, (b) change and (d) relative change are shown. Global mean values are shown on the top right of each figure.**

**Reply for the Anonymous referee #2**

**General Comments:**

A version of the CESM climate model with state-of-the-art new particle formation (NPF) mechanisms is presented, with a focus on the production of Highly Oxygenated organic Molecules (HOMs). The model demonstrates improved agreement with observations. The authors find that organic molecules play a more important role in global NPF than previous studies suggested. Table 5 suggests 83.44% of nucleation proceeds via the mixed $H_2SO_4$-organic pathway below 5.8km, a result that, if nothing else, highlights the importance of further studying this possible NPF pathway.

While not emphasized in the paper, the authors also include an upgraded inorganic NPF mechanism, a potentially very useful innovation.

The article documents a significant effort and it is novel for this level of complexity in new particle formation to be included in a global climate model. The analysis and model evaluation are of high quality with some useful innovations such as the NPF event threshold.

I recommend the paper for publication, subject to responses to the comments below. I also appreciate that, while I do suggest some more sensitivity studies, it is surely not within the scope of the paper to explore all possible uncertainties, as long as the limited nature of the sensitivity studies is properly discussed.

**Response:** We would like to thank the referee for providing the insightful suggestions, which indeed help us further improve the manuscript. We have added required discussion to account for the major and minor comments and marked the corresponding line number in the revised paper. Please see the revision and the response for the comments as follow.

**Major Comment#1**: What are the uncertainties associated with using the Kerminen and Kulmala (2002) approximation for aerosol growth rates up to 20nm in diameter? Many models, including some GCMs that participated in CMIP6, resolve aerosols prognostically and advect them at much smaller sizes. It seems that using the Kerminen and Kulmala (2002) formula in this way for precise studies of NPF might incur quite large biases. Growth rates are not constant with size (Stolzenburg et al, ACP 2020, e.g. Eq. 16), and the more accurate version of the equation that takes better account of the size dependence of the loss rates is given by Lehtinen et al (J. Aerosol Science 2007). I suspect reasonable bounds on these potentially large uncertainties could be determined with some sensitivity studies, without changing the structure of the model.

Also, (L173) should the 0.66 factor be modified between default and updated NPF mechanisms to account for the difference between the 1.7nm starting size for the updated scheme and the 1nm starting size for the default scheme? I believe the parameterization depends on $(1/1.7 - 1/20)$ or $(1/1 - 1/20)$ so this likely makes quite a big difference.

**Response:** We apologize for the typo in Eq. (9) and it is corrected to:

$$ j_{20\text{nm}} = j_{1.7\text{ nm}} \exp\left[-\left(\frac{1}{1.7} - \frac{1}{20}\right)\frac{\gamma\text{CS}}{\text{GR}}\right] \tag{9} $$

However, without changing the structure of the model, we cannot simulate growth rates with size dependence and take account of the size dependence of the loss rates. The main reason is that there is no nucleation mode in CAM6-Chem. For Aitken mode, the geometric diameter is about 20 nm so we cannot resolve aerosols growth rate within 20nm. We have added the assumption and limitation at the **Summary**:

"The NPF rate at about 20nm is calculated based on the Eq. (14) based on Kerminen and Kulmula (2002). It is derived with several simplifying assumptions and approximations:

(1) the only important sink for the newly formed particles is their coagulation to larger preexisting particles;

(2) the newly formed particles grow by condensation at a constant rate;

(3) the preexisting population of larger particles remains unchanged during the newly formed particles growth.

However, Lehtinen et al. (2007) reformulates the previously (Kerminen and Kulmala, 2002) published theory, in which both the accuracy has been improved and the applicability to different conditions is more straightforward. Since CAM6-Chem does not include a nucleation mode, we are not able to

update the NPF rate based on Lehtinen et al. (2007). However, future work, such as implementing a nucleation mode in CAM6-Chem and resolving particle growth rate within 20nm, is worth exploring."


$$N_{>r_p} = N_{mode} \left( \frac{1}{2} + \frac{1}{2} erf \left( \frac{x}{\sqrt{2}} \right) \right) \tag{12}$$

where $x = \ln(r_p/r_m)/\ln\sigma$. $\sigma$ is the geometric standard deviation (the width) of the lognormal distribution $r_m$ is the median radius of the mode. The integral concentration above $r_p$ is therefore $N_{>r_p} = N_{mode} - N_{<r_p}$.

The temporal frequency of the nucleation rate, growth rate, and condensation sink written out of the model are hourly, and the time periods of the model simulation are consistent with the observation period (with an additional month for spin-up). For aerosol number concentrations (including over oceans and land), the model outputs data on a monthly basis, and we compare these monthly averages with observations. When comparing the aerosol and CCN number concentrations with the field campaign in the Amazon basin, the output frequency from model is hourly. Then, we slice the aircraft measurements of aerosol and CCN number concentrations vertical profiles according to the model

output dimensions (4 dimensions including time, height, latitude and longitude). We average all measurements data within each slice and compare it with the corresponding model output data."


Indeed, I would speculate despite the authors' findings, CCN formation in general could still be very sensitive to either the chemical formation of organic molecules (HOMs and ACC) or to the choices made by the modelers on which species participate in NPF. In particular, it is speculative that all HOMs (as defined by the authors) participate in NPF with $H_2SO_4$ at the rate specified in the Riccobono et al (2014) CLOUD study, as in Eq 6 of this paper. The species defined by Riccobono et al was called 'BioOxOrg', not HOM. The large uncertainties associated with these parameterizations should be acknowledged in this paper, and ideally quantified with sensitivity studies.

I would also speculate that the results are likely sensitive to uncertainties in concentrations of $H_2SO_4$ and $NH_3$. The sources of these uncertainties are not really discussed in the main text yet (despite documentation that a bias in $H_2SO_4$ exists and a helpful table of budgets in the supplement).

In the light of these remarks, while I do agree that the uncertainty due to the omission of anthropogenic organics is potentially important, it is far from the only important uncertainty, and the last paragraph of the paper could be more balanced.

**Response:** Yes, the species defined by Riccobono et al. (2014) was called 'BioOxOrg' instead of "HOMs" since at that time the measurement technique was not able to distinguish the nucleating organic species participating in different organic nucleation scheme. We are grateful to the referee for highlighting these important uncertainties and the discussion of both the chemical formation of organic molecules and the choice of nucleating species have already added in new section **"5 Uncertainties from HOMs chemistry"** and the **Summary**. Please refer to the response or major comment#1 of referee #1, #2, #3, #4 and #5 where more details are presented regarding these uncertainties.

Also, uncertainties in concentrations of $H_2SO_4$ and $NH_3$ have added in the **Summary**:

"The overestimation of $H_2SO_4$ in CAM6-Chem could potentially impact our final results regarding the organic proportion in both nucleation and the initial growth rate because both the dependency of inorganic and organic nucleation rate on $H_2SO_4$ concentration are modeled with an exponent greater than 2 (Eq. (2)-(6)). Also, nitrate is not included CMIP6 emissions because of uncertainties in both ammonia emissions and its chemistry and removal (Heald et al., 2012; Paulot et al., 2016). The underestimated nitrate concentrations result in reduced rate of ammonia consumption, potentially leading to an overestimation of residual atmospheric ammonia. Therefore, the inorganic nucleation rate may be overestimated and consequently, the organic proportion of the nucleation rate is likely underestimated."

**Minor Comment#6**: L320 misspelling of species

**Response:** We apologize for the spelling error. We have corrected this.

---

## Author Response (AR2)

**We are very grateful to the evaluations from the reviewers, which have allowed us to clarify and improve the manuscript. Below we addressed the reviewer comments, with the reviewer comments in black and our response in blue or green.**

**Suggestions for revision from Anonymous referee #2:**

The authors responded thoroughly to the review comments. The paper is good and should be highly cited. The scientific content is ready for publication, but ALL of the modifications to the paper need review for written English.

**Response:** We appreciate the positive feedback and note that the scientific content is ready for publication. However, we will ensure that all modifications undergo a thorough review for written English.

The summary and discussion could do with a real conclusion. It is currently a brief summary followed by a long discussion of a list of limitations of the study in a fairly random order (HOMs, then nucleation, then (anthropogenic) HOMs again). This responds to the review comments, but the structure does not really help the readability of the paper or facilitate scientific interpretation of the results. What are the most important uncertainties?

**Response:** Thanks for your insightful feedback regarding the structure of the summary and discussion sections. We have reorganized the sequence of discussions on the uncertainties in this study (Lines 638 to 774). The new order begins with a discussion on uncertainties related to the concentrations of nucleating species, including biogenic HOMs, anthropogenic HOMs and inorganic compounds, followed by the uncertainties in the NPF parameterizations. We have also highlighted that the greatest uncertainty arises from the concentration of biogenic HOMs (Line 638), which is attributed to uncertainties in the chemical reactions. The revised **Summary** is shown below. The Original text is in **Blue**, newly added text is **Underlined**, and the reordered paragraphs are in **Green**.

[revised manuscript text omitted]

I noticed two technical issues in this section: first, nitrate is a secondary species and CMIP6 inventories provide NOx which reacts and partitions to aerosol to make nitrate. So it is not meaningful to comment that CMIP6 inventories do not include nitrate.

**Response:** We are grateful to the referee for pointing out this technical issue. We have revised the manuscript to ensure that the discussion accurately reflects uncertainties in the concentrations of $NH_3$. The following discussion have been included in the **Summary**:

"Ammonia ($NH_3$) emissions used in this study are adapted from the Community Emissions Data System (CEDS), which remain challenging to represent in models due to uncertainties, particularly in specific sectors. $NH_3$ emissions from human waste were adapted using methodologies from the Regional Emissions Inventory in Asia (REAS) (Kurokawa et al., 2013) and rely on a single global default emission factor. Not only is this emission factor uncertain, but there will certainly be regional variations due to differing environmental conditions that we were unable to take into account (Hoesly et al., 2018). For agricultural emissions, the actual practices of managing livestock manures will affect true emissions; such practices vary significantly across the world but are not always well understood or reflected in the emission factors used in global inventories (Paulot et al., 2014). The aforementioned uncertainties in $NH_3$ will affect the inorganic nucleation rate and, consequently, the contribution of organics to the total nucleation rate."


**Table S3**. The ratio of the gas condensation sink (CS) to the aerosol coagulation sink at 1.7 nm (CoagS (1.7 nm)) during NPF events for different m values provided by Lehtinen et al. (2007).

| m values | CS / CoagS (1.7nm) |
|---|---|
| 1.5 | 3.63 |
| 1.6 | 3.98 |
| 1.7 | 4.40 |
| 1.8 | 4.91 |
| 1.9 | 5.51 |

We also added the following discussion (**Underlined**) at the end of the original paragraph (in **Blue**, without underlining) in the **Summary**:

"The NPF rate at around 20 nm is calculated based on Eq. (14) based on Kerminen and Kulmala (2002). It is derived with several simplifying assumptions and approximations: (1) the only important sink for the newly formed particles is their coagulation with larger pre-existing particles; (2) the newly formed particles grow by condensation at a constant rate; (3) the pre-existing population of larger particles remains unchanged during the growth of the newly formed particles. However, Lehtinen et al. (2007) reformulated the previously published theory (Kerminen and Kulmala, 2002) to better account for the size dependence of the loss rates

of newly formed particles (i.e., coagulation sink), rather than simplifying it as the gas condensation sink (CS). The error analysis of the size-dependent coagulation sink (CoagS) compared to the constant CoagS (simplified as gas CS) is shown in Text S5. Recent studies (Stolzenburg et al, 2020; Ozon et al. 2021; Deng et al., 2020) also show that aerosol growth rates are not constant with size. However, CAM6-Chem does not include a nucleation mode, which means that newly formed particles grow from 1.7 nm to 20 nm (geometric diameter in Aitken mode) within one physical timestep (30 minutes), making it impossible to resolve the growth rates of sub-20 nm particles. Future work, such as implementing a nucleation mode in CAM6-Chem and resolving particle growth rates within 20 nm, is worth exploring."